# Probability-Entropy Calibration: An Elastic Indicator for Adaptive Fine-tuning

**Wenhao Yu** [1]  **Shaohang Wei** [2]  **Jiahong Liu** [1]  **Yifan Li** [1]  **Minda Hu** [1]  **Aiwei Liu** [3]  **Hao Zhang** [4]  **Irwin King** [1]

## Abstract

Token-level reweighting is a simple yet effective mechanism for controlling supervised fine-tuning, but common indicators are largely one-dimensional: the ground-truth probability reflects downstream alignment, while token entropy reflects intrinsic uncertainty induced by the pre-training prior. Ignoring entropy can misidentify noisy or easily replaceable tokens as learning-critical, while ignoring probability fails to reflect target-specific alignment. RANKTUNER introduces a probability–entropy calibration signal, the *Relative Rank Indicator*, which compares the rank of the ground-truth token with its expected rank under the prediction distribution. The inverse indicator is used as a token-wise *Relative Scale* to reweight the fine-tuning objective, focusing updates on truly under-learned tokens without over-penalizing intrinsically uncertain positions. Experiments on multiple backbones show consistent improvements on mathematical reasoning benchmarks, transfer gains on out-of-distribution reasoning, and pre code generation performance over probability- or entropy-only reweighting baselines. The implementation code is available at `https://github.com/LvAoAo/Ranktuner_VERL`.

## 1. Introduction

With the remarkable progress of large language models (LLMs) in natural language processing (Brown et al., 2020; Devlin et al., 2019), mathematical reasoning (Cobbe et al., 2021; Hendrycks et al., 2021), code generation (Chen et al., 2021; Wang et al., 2021), and knowledge retrieval (Lewis et al., 2020), finetuning has become a pivotal technique (Kumar et al., 2025; Tie et al., 2025) for efficiently adapting

the general capabilities of LLMs to specific downstream tasks. By selectively updating models with datasets of varying sizes and domains, finetuning significantly enhances accuracy, robustness, and practicality in targeted application areas, while also fulfilling requirements for safety and preference alignment (Rafailov et al., 2023). Finetuning methodologies encompass a variety of paradigms such as supervised finetuning (SFT) (Hu et al., 2022; Houlsby et al., 2019) and reinforcement learning (RL) (Schulman et al., 2017), and have been widely adopted in academic and industrial settings across multilingual (Liu et al., 2020; Lample & Conneau, 2019), multitask (Liu et al., 2019), and cross-modal scenarios (Radford et al., 2021; Ramesh et al., 2021).

Recent token-level reweighting methods for fine-tuning can be broadly categorized into two paradigms based on the statistics they rely on: *Prob-dominant* weighting that designs $w_t$ as a function of the ground-truth probability $p_t$ (Liu et al., 2025b; Wu et al., 2026; Lin et al., 2026), and *Entropy-dominant* weighting that uses the predictive uncertainty $H_t$ as the reweighting signal (Diao et al., 2026).

While effective in specific settings, prior work typically treats $p_t$ and $H_t$ in isolation, which can misidentify *what to emphasize*. **Noisy tokens** often fall into a "noise region" where neither one-dimensional signal is reliable: entropy-dominant schemes can up-weight them for high uncertainty, while prob-dominant schemes may overreact to atypical but irrelevant tokens. A controlled noise-insertion diagnostic shows both baselines surface injected noise far more than our indicator, as seen in Tab. 1 (detailed in App. B.3). **Replaceable tokens** (e.g., "essentially" vs. "basically") are intrinsically ambiguous and high-entropy, so a low $p_t$ need not indicate a true error—making prob-dominant weighting overly sensitive and potentially harmful to linguistic flexibility. These regimes motivate a *calibrated* token-importance signal that contextualizes likelihood by intrinsic uncertainty, down-weighting noisy/replaceable tokens while focusing updates on genuinely critical failures.

Motivated by these findings, we propose *RankTuner*, which calibrates *downstream alignment* by *intrinsic uncertainty*. Our contributions are threefold:

1. We analyze *Prob-Dominant* vs. *Entropy-Dominant* token reweighting and show why one-dimensional weighting can over-emphasize *noisy* and *replaceable*

[1]The Chinese University of Hong Kong [2]Peking University [3]Tsinghua University [4]University of the Chinese Academy of Sciences. Correspondence to: Wenhao Yu <yuwenhao117@gmail.com>, Hao Zhang <zh.cs.star@gmail.com>.

*Proceedings of the 43rd International Conference on Machine Learning*, Seoul, South Korea. PMLR 306, 2026. Copyright 2026 by the author(s).

*Table 1.* **Noise sensitivity.** Token noise precision/recall@10% and sequence noise hit@10% (App. B.3).

| METHOD | TOK PREC@10% ($\downarrow$) | TOK REC@10% ($\downarrow$) | SEQ HIT@10% ($\downarrow$) |
|---|---|---|---|
| ENTROPY-DOMINANT | 4.54% | 55.33% | 77% |
| PROB-DOMINANT | 3.25% | 39.65% | 77% |
| RANKTUNER (OURS) | **2.16**% | **26.39**% | **9**% |

tokens, since $p_t$ (alignment) and $H_t$ (intrinsic uncertainty) capture different factors (Sec. 3).

2. We introduce a rank-based view with a *Relative Rank* signal that compares the target-token rank against its expected rank, yielding an uncertainty-aware adaptive token reweighting scheme for fine-tuning (Sec. 4).

3. We validate RankTuner across base models and reasoning benchmarks, with ablations supporting the complementary roles of probability- and entropy-aware components (Sec. 5).

## 2. Preliminaries

In this section, we establish the formal notation and theoretical foundation for our method. We first introduce a unified weighting framework for fine-tuning, followed by the core concepts of model uncertainty and the guessing problem.

### 2.1. Problem Formulation and Unified Weighting

Consider a dataset $\mathcal{D}$ consisting of prompt-response pairs $(x, y)$, where $x$ is the input prompt and $y = (y_1, y_2, \ldots, y_T)$ is the target response of length $T$. Let $\pi_\theta$ denote a language model parameterized by $\theta$. At each decoding step $t \in \{1, \ldots, T\}$, the model generates a probability distribution over the vocabulary $\mathcal{V}$. We denote the probability assigned to the $i$-th token $v_i \in \mathcal{V}$ as $p_{t,i} = \pi_\theta(v_i \mid y_{<t}, x)$. The probability of the actual ground-truth token $y_t$ in the sequence is denoted as $p_t = \pi_\theta(y_t \mid y_{<t}, x)$.

Many fine-tuning objectives can be formulated as minimizing a weighted negative log-likelihood (NLL) loss:

$$\mathcal{L}(\theta) = \mathbb{E}_{(x,y) \sim \mathcal{D}} \left[ -\sum_{t=1}^{T} w_t \log p_t \right].$$

The weighting coefficient $w_t$ determines the relative importance of each token during optimization.

**Supervised Fine-tuning (SFT).** Standard SFT treats every token in the target sequence as equally informative, assigning a uniform weight $w_t = 1$. This approach does not account for the varying difficulty or information density across different parts of the response.

### 2.2. Token Entropy

Token entropy measures the uncertainty of the model's prediction at step $t$. Using the vocabulary-wide probabilities $p_{t,i}$ defined earlier, the entropy $H_t$ is:

$$H_t = -\sum_{i=1}^{|\mathcal{V}|} p_{t,i} \log p_{t,i}.$$

A high $H_t$ indicates a flat distribution, while a low $H_t$ indicates a sharp distribution.

### 2.3. The Guessing Problem

The Guessing Problem formulates the challenge of identifying the realization of a discrete random variable $X$ through a sequence of queries (Massey, 1994). Specifically, in a sequential guessing setting, one asks "Is $X$ equal to $x_i$?" for candidate values $x_i$ until the answer is affirmative. Let $G$ denote the number of guesses required. The optimal strategy to minimize the expected number of guesses, $\mathbb{E}[G]$, is to query values in descending order of their probabilities. If the probability distribution $\mathbf{p}$ is sorted such that $p_{\hat{1}} \geq p_{\hat{2}} \geq \ldots$, where $\hat{i}$ denotes the rank index, the minimum expected number of guesses is given by:

$$\mathbb{E}[G] = \sum_{\hat{i}=1}^{\infty} \hat{i} \cdot p_{\hat{i}}.$$

This quantity reflects the effective number of candidates one must examine to find the target, serving as an intuitive measure of uncertainty.

## 3. Token Reweighting Paradigms: A Joint View of Probability and Entropy

In this section, we first provide a formal characterization of two dominant token reweighting paradigms (Sec. 3.1), and then develop a more systematic analysis of why single-dimensional weighting schemes are fundamentally inadequate (Sec. 3.2).

### 3.1. *Prob-Dominant* and *Entropy-Dominant* Importance Weighting $w_t$

We categorize existing importance weighting strategies into two primary paradigms based on the statistical properties of token predictions: **Prob-Dominant** and **Entropy-**

**Dominant**. These paradigms differ fundamentally in which aspect of the predictive distribution they emphasize.

**Prob-Dominant Weighting.** The *Prob-Dominant* weighting methods primarily rely on the ground-truth probability $p_t$ as the token-wise signal for reweighting. Intuitively, $p_t$ reflects how much probability mass the model assigns to the labeled continuation at step $t$, and thus serves as a direct proxy for token-level task alignment. Accordingly, the importance weight is parameterized as a function of $p_t$. Depending on the objective, $\phi_{\text{prob}}(\cdot)$ may take either an increasing or a decreasing form: $w_t^{\text{prob}} = \phi_{\text{prob}}(p_t)$.

**Entropy-Dominant Weighting.** The *Entropy-Dominant* paradigm focuses on the global uncertainty of the predictive distribution. High entropy $H_t$ signifies that the probability mass is dispersed across multiple candidates, reflecting ambiguity in the model's decision boundary. Consequently, the importance weight $w_t$ is designed to be *positively correlated* with $H_t$, increasing the fine-tuning signal for tokens exhibiting *high uncertainty*. Formally, $w_t$ can be defined as a *monotonically increasing* function of the predictive entropy $H_t$: $w_t^{\text{ent}} = \phi_{\text{ent}}(H_t)$.

### 3.2. Deeper Insights into the Different Finetuning Paradigms

We argue that the limitation of existing paradigms stems from treating Token Entropy and Ground-Truth Probability in isolation. A unified view reveals that they capture orthogonal aspects of the generation process: Fig. 1 provides an intuition for why meaningful fine-tuning signals should depend on both dimensions jointly.

**Entropy ($H_t$) as Intrinsic Linguistic Uncertainty.** Entropy is a functional of the entire distribution $\pi_\theta(\cdot \mid x_{\leq t})$, independent of the specific ground-truth token $y_t$. It reflects the model's *intrinsic uncertainty* about what could plausibly come next under its pre-training prior. High entropy typically appears at positions that admit many valid continuations (e.g., open-ended reasoning or descriptive phrasing), while low entropy corresponds to more deterministic roles (e.g., syntax, fixed formats). In this sense, $H_t$ provides a *prior* on the difficulty/ambiguity of the position.

**Probability ($p_t$) as Downstream Task Alignment.** The ground-truth probability $p_t = \pi_\theta(y_t \mid x_{\leq t})$ quantifies how much the model supports the *specific* labeled continuation and thus serves as a token-level proxy for downstream task alignment. Unlike entropy, $p_t$ is target-specific: it directly determines the supervised pressure to increase mass on $y_t$, indicating where the model is misaligned with the task objective.

**The Pitfall of One-Dimensional Weighting.** Consider the ground-truth token sequence in Fig. 1 (left), where we study two cases that share the same prefix **"The answer *is*, *umm***

***essentially*"** but differ in the final token: one with ground-truth answer **5** and one with ground-truth answer **6**. Each token exhibits distinct $(p_t, H_t)$ characteristics, exposing the brittleness of single-dimensional approaches.

- *Entropy-Dominant* methods assign high importance to tokens like "*umm*" simply because they induce high entropy—yet such filler words are inherently noisy and contribute little semantic value. Up-weighting them amplifies uninformative gradients and may degrade alignment with the downstream task.

- *Prob-Dominant* methods heavily penalize any token with low $p_t$, including positions like "*essentially*" where multiple synonyms (e.g., "basically", "roughly") are equally acceptable. Over-correcting such naturally ambiguous choices risks distorting the model's pre-trained linguistic flexibility.

These observations suggest that an effective weighting scheme must evaluate $p_t$ *in the context of* the underlying uncertainty $H_t$: it should down-weight high-entropy, easily replaceable tokens while maintaining strong emphasis on low-entropy positions where mistakes correspond to critical failures. This joint treatment naturally mitigates both noise amplification and the over-penalization of semantically flexible words. As illustrated in Fig. 1 (right), tokens across the $(p_t, H_t)$ space fall into four qualitatively distinct regimes (①–④) that cannot be uniformly handled by one-dimensional paradigms. Our *Relative Rank Indicator* $I_t$ operationalizes this joint perspective by coupling both dimensions into a unified weighting framework: the background color visualizes $I_t$ values over the $(p_t, H_t)$ plane, discriminatively delineating these regimes and revealing a central "Noise Region" (⑤) where high-entropy tokens receive appropriately low emphasis. A concrete token-level example in App. B.4 (Fig. 5) shows the same pattern: relatively replaceable or noisy tokens are down-weighted, while genuinely incorrect result-critical tokens receive stronger emphasis.

## 4. Methodology: Bridging the Gap via Rank-Based Discretization

This section tackles a key challenge highlighted in Sec. 3: how to combine ground-truth probability $p_t$ and token entropy $H_t$ into a single, principled token-wise scaling signal for fine-tuning. To make $p_t$ and $H_t$ *comparable*, we develop a rank-based discretization grounded in the rank statistics and a guessing view. Specifically, we define the *Relative Rank Indicator* $\mathcal{I}_t$ from $(R_t, \mathbb{E}[R_t])$ (Sec. 4.1), formalize *relative competence* $C_t = \rho(p_t)/\kappa(H_t)$ as a calibration target (Sec. 4.2), establish tight bounds linking $(p_t, H_t)$ to $(R_t, \mathbb{E}[R_t])$ (Sec. 4.3), and derive a concrete calibra-

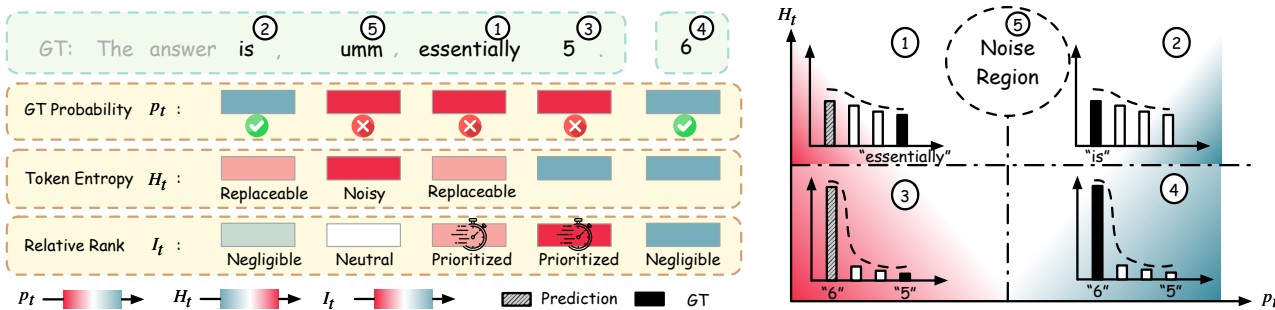

Figure 1. **A joint view of token correctness and intrinsic uncertainty.** **(Left)** Token-level visualization of three indicators: the ground-truth probability $p_t$, token entropy $H_t$, and our Relative Rank Indicator $I_t$ (Sec. 4). Colors encode relative magnitude; arrows indicate the increasing direction. **(Right)** A schematic in the $(p_t, H_t)$ plane with four regimes (①–④) distinguished by $I_t$; the background color gradient encodes $I_t$ values; inset histograms show representative predictive distributions for typical tokens (e.g., "essentially", "is", "5", "6"); the dashed circle marks a *Noise Region* (⑤).

---

tion $(\widehat{\rho}, \widehat{\kappa})$ via the Cauchy Mean Value Theorem (CMVT) (Sec. 4.4). **Building on these analyses, we introduce the *Relative Scale* $\mathcal{S}_t = \mathcal{I}_t^{-1}$ and integrate it into fine-tuning objectives (Sec. 4.5).**

### 4.1. Relative Rank Indicator

We start from the guessing view (Sec. 2.3) and define a token-level indicator that compares the realized outcome to the model's intrinsic uncertainty at the same position.

**Definition 4.1** (Rank and Expected Rank). At decoding step $t$, let $R_t$ denote the rank of the ground-truth token $y_t$ when candidates are sorted by decreasing probability. We define the *Expected Rank* as the guessing cost of sampling from the model distribution:

$$\mathbb{E}[R_t] = \sum_{\hat{i}=1}^{|\mathcal{V}|} \hat{i} \cdot p_{t,\hat{i}}, \qquad (1)$$

where $p_{t,\hat{i}}$ is the $\hat{i}$-th largest probability in the distribution.

**Definition 4.2** (Relative Rank Indicator). We define the *Relative Rank Indicator* $\mathcal{I}_t$ as

$$\mathcal{I}_t = g\left(f(R_t) - f(\mathbb{E}[R_t])\right), \qquad (2)$$

where $f(x)$ is a *monotonically decreasing* **transformation function** and $g(x)$ is a *monotonically increasing* **scaling function**. In our proposed framework, we specifically instantiate these functions as

$$f(x) = \frac{1}{\log_2(x+1)}, \quad g(x) = 2^x.$$

The choice of $f$ follows a logarithmic decay strategy common in ranking metrics (Järvelin & Kekäläinen, 2017), and $g$ normalizes the neutral case to $\mathcal{I}_t = 1$ when $R_t = \mathbb{E}[R_t]$. We emphasize that the key signal is the *relative discrepancy* between $R_t$ and $\mathbb{E}[R_t]$; the particular $(f, g)$ is chosen

for stability and a closed form, not claimed optimal. Alternative monotone choices lead to broadly stable results (App. C.5), suggesting that the main gain comes from the rank-based probability–entropy calibration principle rather than a specific handcrafted transformation pair.

Fig. 2 (left) visualizes the behavior of the Relative Rank Indicator. As shown, $\mathcal{I}$ decreases with larger Rank $R$ (lower accuracy) but increases with larger Expected Rank $\mathbb{E}[R]$ (higher difficulty). This dynamic ensures that the model receives greater rewards for correct predictions in high-uncertainty contexts compared to simple scenarios, effectively balancing performance evaluation with task difficulty. Moreover, due to the logarithmic compression in $f(\cdot)$ and the exponential rescaling in $g(\cdot)$, $\mathcal{I}$ rapidly saturates around 1 once both $R$ and $\mathbb{E}[R]$ become sufficiently large, yielding an approximately *neutral regime* where differences in low-likelihood tokens are deemphasized due to the high uncertainty.

Beyond the theoretical surface, we also visualize the empirical distributions of $(R, \mathbb{E}[R], \mathcal{I})$ on chain-of-thought tokens. We observe that $\mathbb{E}[R]$ is typically small, while $R$ is heavy-tailed. Crucially, $\mathcal{I}$ effectively separates different token types: replaceable pronouns such as "them" and "all" (highlighted as red triangles) reside in the neutral region ($\mathcal{I} \approx 1$) where substitutable tokens yield little signal, whereas critical computation tokens like the fraction operator "frac", key result "0", and delimiter "{" (yellow triangles) concentrate in the low-$\mathcal{I}$ region (deep red zone, $\mathcal{I} < 1$), indicating high sensitivity to prediction accuracy. This clear separation indicates that $\mathcal{I}$ cleanly differentiates *high-uncertainty errors* from *low-uncertainty yet wrong* predictions, by contrasting the realized rank $R$ against the context-conditioned expected rank $\mathbb{E}[R]$.

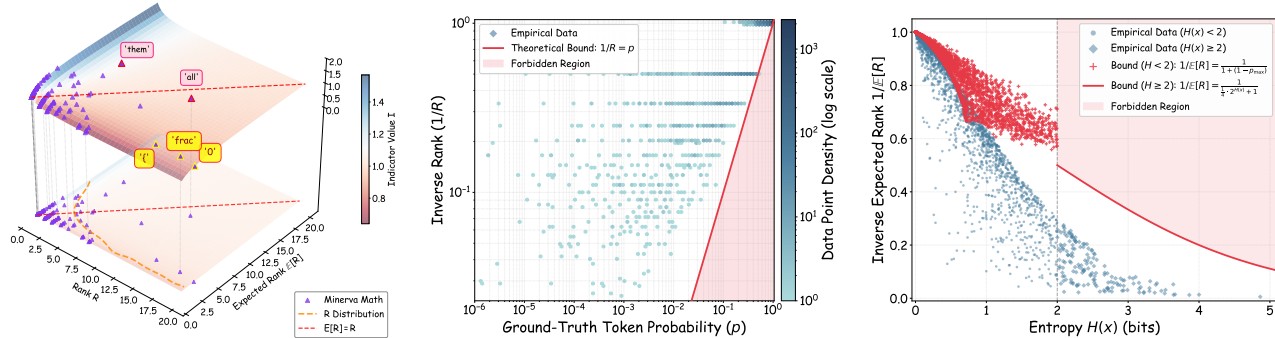

*Figure 2.* **Visualization and empirical validation of rank-based metrics on Qwen3-8B predicted chain-of-thought tokens from the Minerva Math dataset. (Left)** 3D visualization of the Relative Rank Indicator $\mathcal{I}$ as a function of Rank $R$ and Expected Rank $\mathbb{E}[R]$. The indicator incentivizes accurate predictions (low $R$) specifically in difficult contexts (high $\mathbb{E}[R]$). **(Middle)** Rank $R$ vs. probability $p$, showing adherence to the upper bound $R \leq 1/p$ (Eq. (4)). **(Right)** Expected rank $\mathbb{E}[R]$ vs. entropy $H$, demonstrating alignment with the lower bound in Eq. (5). Note that the subscript $t$ is omitted here as we represent aggregate statistics over all tokens.

## 4.2. Relative Competence Template

As discussed in Sec. 3, the ground-truth probability $p_t$ measures *upstream-to-downstream alignment*, while the predictive entropy $H_t$ summarizes uncertainty from the pretraining prior. We therefore assess alignment *conditional on* prior support: how well the model explains the target token *given* the context. This mirrors the conditional-probability template: $\Pr(A \mid U) = \frac{\Pr(A,U)}{\Pr(U)}$, where $A$ is the downstream *alignment event* and $U$ is the upstream *prior-support event*. In our token-level setting, we treat $p_t$ as a proxy for the joint term $\Pr(A,U)$, and map $H_t$ to an effective support term $\Pr(U)$: higher entropy means the predictive mass is more diffuse and thus provides weaker support for a sharp prediction.

**Definition 4.3** (Relative Competence Template). Motivated by this analogy, we introduce an abstract token-level *relative competence* score

$$C_t \triangleq \frac{\rho(p_t)}{\kappa(H_t)}, \tag{3}$$

where $\rho(\cdot)$ is a monotonically increasing function of $p_t$, and $\kappa(\cdot)$ maps entropy to an *effective prior-support term* and is therefore taken to be monotonically *decreasing* in $H_t$ (high uncertainty $\Rightarrow$ weaker prior support). Under this semantics, a *small* $C_t$ indicates that the model is insufficiently aligned *relative to* the context difficulty, whereas a *large* $C_t$ suggests the position is already well-explained and can be downweighted. The technical question then becomes how to choose or approximate $\rho$ and $\kappa$ in a principled way.

## 4.3. Bridging Bounds Between $(p_t, H_t)$ and $(R_t, \mathbb{E}[R_t])$

Our key insight is that the realized rank $R_t$ and the expected rank $\mathbb{E}[R_t]$ provide a natural bridge: both quantify guessing cost and are therefore directly *comparable*, whereas $p_t$ and

$H_t$ lack such a direct connection. Moreover, they admit tight, complementary bounds: $R_t$ is upper bounded by $1/p_t$, while $\mathbb{E}[R_t]$ is lower bounded by a function of $H_t$.

**Proposition 4.4** (Rank–Probability Bound). *Let the probability distribution at position $t$ be sorted such that $p_{t,\hat{1}} \geq p_{t,\hat{2}} \geq \cdots$. For the ground-truth token with probability $p_t$ and rank $R_t$, we have*

$$R_t \leq \frac{1}{p_t}. \tag{4}$$

A proof is provided in App. A.1.

**Proposition 4.5** (Expected Rank–Entropy Bound). *The expected rank $\mathbb{E}[R_t]$ is lower bounded by a function of entropy:*

$$\mathbb{E}[R_t] \geq \begin{cases} \frac{1}{4}\, 2^{H_t} + 1, & \text{if } H_t \geq 2, \\ 2 - p_{\max,t}, & \text{if } H_t < 2, \end{cases} \tag{5}$$

where $p_{\max,t}$ denotes the maximum probability in the distribution at position $t$. A proof is deferred to App. A.2.

**Empirical Validation of Bounds.** To validate these theoretical bounds and their tightness in practice, we visualize the relationships on chain-of-thought tokens from the Minerva Math dataset (Lewkowycz et al., 2022), as predicted by Qwen3-8B (Yang et al., 2025). As shown in the middle and right panels of Fig. 2, the plot of $R$ against $p$ closely follows the upper envelope $R = 1/p$ from Eq. (4), while $\mathbb{E}[R]$ versus $H$ aligns well with the lower bound from Eq. (5). In both cases, the empirical distributions closely adhere to the predicted boundaries, confirming that rank-based quantities can effectively serve as discrete, commensurate proxies for probability and entropy, respectively. Fig. 3 further shows that the inverse expected-rank gap is concentrated near zero.

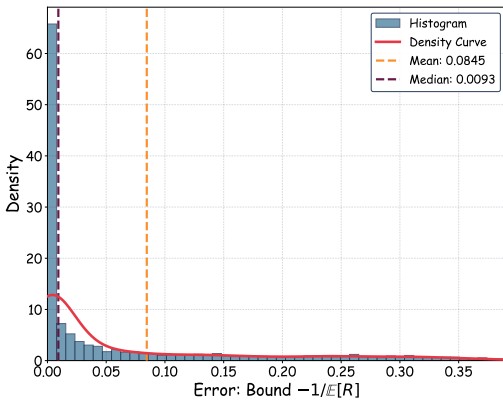

*Figure 3.* **Error distribution for the expected-rank bound on Qwen3-8B (Minerva Math, tokens 0–29).** The plot shows $\frac{1}{s(H)} - \frac{1}{\mathbb{E}[R]}$, where $s(H)$ is the entropy-based lower bound in Eq. (5). The mass near zero indicates that the entropy-induced bound closely tracks $1/\mathbb{E}[R]$.

See App. B.5 for the complementary rank–probability gap and summary statistics.

## 4.4. Deriving $\widehat{\rho}$ and $\widehat{\kappa}$ via CMVT

Having established tight bounds connecting rank-based quantities to probability and entropy, we now instantiate the functions $\rho(p_t)$ and $\kappa(H_t)$ by leveraging the Relative Rank Indicator $\mathcal{I}_t$ introduced in Eq. (2).

**Connecting $\mathcal{I}_t$ to competence via the Cauchy Mean Value Theorem.** Recall that $\mathcal{I}_t = 2^{f(R_t) - f(\mathbb{E}[R_t])}$ with $f(x) = \frac{1}{\log_2(x+1)}$. To express $f(R_t) - f(\mathbb{E}[R_t])$ in a log-ratio form compatible with the competence ratio, we apply CMVT to $f$ with the auxiliary function $v(x) = \log_2 x$. By the Cauchy Mean Value Theorem, there exists an intermediate value $\xi_t$ strictly between $R_t$ and $\mathbb{E}[R_t]$ such that (see App. A.3):

$$f(R_t) - f(\mathbb{E}[R_t]) = -K(\xi_t) \cdot (\log_2 R_t - \log_2 \mathbb{E}[R_t]), \tag{6}$$

where $K(\xi_t) = \frac{\xi_t}{(\xi_t+1)[\log_2(\xi_t+1)]^2} > 0$. Consequently, $\mathcal{I}_t$ admits a power-law form that directly connects it to the competence ratio:

$$\mathcal{I}_t = 2^{-K(\xi_t) \cdot \log_2(R_t/\mathbb{E}[R_t])} = \left(\frac{\mathbb{E}[R_t]}{R_t}\right)^{K(\xi_t)}. \tag{7}$$

For typical reasoning tokens where both $R_t$ and $\mathbb{E}[R_t]$ are small, we have $K(\xi_t) \approx 0.5$ (see App. A.4 for analysis).

**Constructing rank-based surrogates $\widehat{\rho}$ and $\widehat{\kappa}$.** To operationalize $C_t$ in terms of rank-based quantities, we define surrogates $\widehat{\rho}(p_t)$ and $\widehat{\kappa}(H_t)$ by directly exploiting the established bounds together with the coefficient $K(\xi_t)$ from the Cauchy analysis. From Eq. (4) (which gives $R_t \lesssim 1/p_t$) and the power-law structure revealed in Eq. (7), we set

$\widehat{\rho}(p_t) \triangleq p_t^{K(\xi_t)}$ as a proxy for $R_t^{-K(\xi_t)}$. Similarly, letting $s(H_t)$ denote the right-hand side of Eq. (5) (a lower bound for $\mathbb{E}[R_t]$), we set $\widehat{\kappa}(H_t) \triangleq s(H_t)^{-K(\xi_t)}$ as a proxy for $\mathbb{E}[R_t]^{-K(\xi_t)}$. With these definitions, the surrogate ratio

$$\widehat{C}_t \triangleq \frac{\widehat{\rho}(p_t)}{\widehat{\kappa}(H_t)} = \left(p_t \, s(H_t)\right)^{K(\xi_t)} \tag{8}$$

approximates the competence ratio $\left(\frac{\mathbb{E}[R_t]}{R_t}\right)^{K(\xi_t)}$ by substituting the rank-based bounds into the power-law form.

**Relating $\mathcal{I}_t$ to the competence score.** Combining Eq. (7) with the surrogate construction above, we observe that the Relative Rank Indicator $\mathcal{I}_t$ directly approximates the competence ratio:

$$\mathcal{I}_t = \left(\frac{\mathbb{E}[R_t]}{R_t}\right)^{K(\xi_t)} \gtrsim \left(\frac{s(H_t)}{1/p_t}\right)^{K(\xi_t)} = \widehat{C}_t. \tag{9}$$

In this manner, our approach bridges probability $p_t$ and entropy $H_t$ through a unified rank-based framework. The functions $\widehat{\rho}$ and $\widehat{\kappa}$ represent one concrete instantiation of the general template $C_t = \rho(p_t)/\kappa(H_t)$, where the rank-to-probability/entropy correspondences are given by the theoretical bounds (Eqs. (4) and (5)). This construction is validated by the empirical adherence observed in Fig. 2. App. A.5 further justifies the surrogate substitution and establishes the boundedness/tightness guarantees. **The resulting formulation enables practical application of competence-aware weighting in supervised fine-tuning, as we detail in the following subsection.**

## 4.5. Implementation of Relative-Rank Guided Losses

The Relative Rank Indicator $\mathcal{I}_t$ measures token-level performance relative to uncertainty. For fine-tuning, we focus on its inverse: assigning larger weights to tokens where the model underperforms relative to expectation, while down-weighting already well-mastered tokens to avoid over-optimization.

We term this weighting signal the **Relative Scale** ($\mathcal{S}_t$):

$$\mathcal{S}_t \triangleq \mathcal{I}_t^{-1} \approx \left(p_t \cdot s(H_t)\right)^{-K(\xi_t)},$$
$$\xi_t := \max\{R_t, \, s(H_t)\}, \tag{10}$$
$$K(\xi_t) := \left[\log_2(\xi_t + 1)\right]^{-2}.$$

For simplicity and training stability, we omit the multiplicative factor $\frac{\xi_t}{\xi_t+1}$ in $K(\xi_t)$. Other approximations of $\xi$ are discussed in App. C.6. We incorporate the Relative Scale $\mathcal{S}_t$ into supervised fine-tuning by modulating the token-level weighting coefficients. Following the unified formulation in Sec. 2, we replace the original weight $w_t$ with a variant:

$$\widetilde{w}_t = w_t \cdot \mathcal{S}_t. \tag{11}$$

Algorithm 1 provides the pseudocode for this procedure. Practically, we set $w_t = p_t$ for all fine-tuning tasks on math reasoning datasets, and $w_t = 1$ for general fine-tuning tasks; the rationale is provided in App. B.6.

**Relation to RL-style post-training.** RankTuner is complementary to RL-style post-training methods such as PPO (Schulman et al., 2017) and GRPO (Shao et al., 2024). In RL post-training, entropy often serves as an exploration regularizer. RankTuner does not reward entropy directly; it uses entropy to contextualize token difficulty relative to the model's expected rank. A natural but currently unvalidated extension is to inject the Relative Scale into PPO/GRPO-style token-level policy ratios, or to use relative rank and entropy for token selection; App. B.7 gives one illustrative form, and we leave these RL variants for future work.

# 5. Experiments

We design experiments to answer the following questions: **(RQ1)** *Effectiveness*: Does RANKTUNER consistently improve mathematical reasoning performance over the original models and representative probability-/entropy-based fine-tuning baselines across benchmarks and decoding budgets (Pass@1/Pass@16)? **(RQ2)** *Out-of-distribution generalization*: Does RANKTUNER generalize beyond mathematical reasoning to diverse reasoning benchmarks? **(RQ3)** *Key ingredients*: How do the probability-aware and entropy-aware components contribute to the gains, and how does RANKTUNER compare to loss-shaping alternatives?

## 5.1. Experimental Setup

Following prior work (Wu et al., 2026), we train on the NuminaMath-CoT dataset (Jia et al., 2024) using the first 10k training instances. We run experiments with multiple base models, including Qwen2.5-Math-7B (Yang et al., 2024) and Qwen3-8B (Yang et al., 2025). We further report supplementary cross-architecture results in the Tab. 9.

**Implementation Details.** Our implementation is built on the `verl` framework (Sheng et al., 2025). All experiments can be completed on four NVIDIA A800-SXM4-80GB GPUs. We use the AdamW optimizer with a learning rate of $5 \times 10^{-5}$ for all models. We set the global mini-batch size to 256 and the maximum input length to 2048 tokens. The learning rate follows a cosine decay schedule with a warm-up ratio of 0.1.

For evaluation, we generate 16 decoding runs with temperature 1.0 and maximum generation length of 4096 tokens, and report Pass@1 and Pass@16 (see App. C.3 for the Pass@$k$ definition). We evaluate on Math500 (Lightman et al., 2024), Minerva Math (Lewkowycz et al., 2022), OlympiadBench (He et al., 2024), AIME 2024, and AMC 2023.

**Baselines and Metrics.** We compare RANKTUNER against standard SFT and representative token-level loss reweighting baselines. In particular, OverTone, DFT, and TALR are *probability-dominant* weighting schemes driven primarily by the ground-truth token probability $p_t$ (possibly with gating/temperature), while EAFT is an *entropy-dominant* scheme that weights tokens based on (top-$K$) predictive entropy (see App. C.2 for more detailed comparisons of the baselines). We report Pass@$k$ (mainly Pass@1 and Pass@16), i.e., the probability that at least one out of $k$ sampled solutions is correct (see App. C.3 for the Pass@$k$ definition and computation).

## 5.2. RQ1: Effectiveness on Reasoning Tasks

Tab. 2 compares RANKTUNER with representative probability- and entropy-based fine-tuning baselines across five mathematical reasoning benchmarks, reporting gains both over the original model and over the strongest competing fine-tuning baseline. Across both backbones, RANKTUNER delivers consistent improvements over the original models, with particularly strong gains in Pass@1 on MATH-OAI, Minerva Math, and OlympiadBench; meanwhile, it also boosts Pass@16 on most benchmarks, indicating that the improved single-sample accuracy does not come at the expense of multi-sample coverage. Notably, on **AIME24**—a comparatively hard benchmark where several baselines exhibit substantial degradation (e.g., reduced Pass@16 and/or Pass@1)—RANKTUNER *maintains* the original Pass@16 while still improving Pass@1, suggesting a more robust calibration of learning signals that avoids over-correcting intrinsically uncertain positions. We observe one mild trade-off on Qwen2.5-Math-7B for **AMC23** Pass@16, which decreases slightly despite a large Pass@1 gain; overall, the $\Delta_{\text{Best}}$ rows show that RANKTUNER achieves the best or near-best performance across the majority of benchmark–metric pairs.

## 5.3. RQ2: Out-of-Distribution Generalization

To evaluate the generalization capability of RANKTUNER beyond mathematical reasoning, we conduct experiments on two diverse reasoning benchmarks: ARC-C (Clark et al., 2018) and GPQA (Rein et al., 2024). Following our experimental protocol, we set the sampling temperature to 0.8, generate 16 candidate responses per query, and use a maximum token budget of 3072 to accommodate comprehensive reasoning traces. Tab. 3 summarizes the Pass@1 performance of various methods on Qwen2.5-Math-7B.

Overall, Tab. 3 shows that RANKTUNER achieves the best on ARC-C and GPQA, indicating robust out-of-distribution transfer beyond math reasoning. In contrast, DFT is a probability-dominant reweighting method that prioritizes already-confident tokens, which can induce over-sharpening

*Table 2.* Performance comparison on mathematical reasoning benchmarks. We report Pass@1 and Pass@16 metrics. Best results for each base model are in bold. $\Delta_{\text{Orig}}$ denotes RANKTUNER minus the Original base model, while $\Delta_{\text{Best}}$ denotes RANKTUNER minus the best non-RANKTUNER fine-tuning baseline in the same block.

| Model | Method | MATH-OAI | | Minerva Math | | OlympiadBench | | AIME24 | | AMC23 | |
|---|---|---|---|---|---|---|---|---|---|---|---|
| | | P@1 | P@16 | P@1 | P@16 | P@1 | P@16 | P@1 | P@16 | P@1 | P@16 |
| Qwen2.5-Math-7B | Original | 31.79 | 87.80 | 7.63 | 42.28 | 9.49 | 47.85 | 6.25 | **23.33** | 20.47 | **85.00** |
| | SFT | 53.52 | 88.20 | 17.74 | 50.00 | 19.14 | 55.11 | 2.08 | 10.00 | 24.53 | 82.50 |
| | EAFT | 53.94 | 87.40 | 20.24 | 59.19 | 18.96 | 54.37 | 2.50 | 10.00 | 24.22 | 67.50 |
| | OverTone | 47.00 | 87.80 | 18.89 | 51.47 | 16.02 | 51.56 | 2.50 | 20.00 | 25.16 | 75.00 |
| | DFT | **69.15** | 85.00 | 26.06 | 40.07 | 32.62 | 54.81 | 4.17 | 16.67 | 41.09 | 72.50 |
| | TALR | 68.83 | 87.40 | **35.68** | **60.29** | 32.87 | 57.78 | 6.67 | 16.67 | 43.44 | 77.50 |
| | RANKTUNER | 68.60 | **88.80** | 33.30 | 59.56 | **32.89** | **62.07** | **7.08** | **23.33** | 44.53 | 82.50 |
| | $\Delta_{\text{Orig}}$ | ↑ 36.81 | ↑ 1.00 | ↑ 25.67 | ↑ 17.28 | ↑ 23.40 | ↑ 14.22 | ↑ 0.83 | ↑ 0.00 | ↑ 24.06 | ↓ 2.50 |
| | $\Delta_{\text{Best}}$ | ↓ 0.55 | ↑ 0.60 | ↓ 2.38 | ↓ 0.73 | ↑ 0.02 | ↑ 4.29 | ↑ 0.41 | ↑ 3.33 | ↑ 1.09 | ↑ 0.00 |
| Qwen3-8B | Original | 65.14 | 87.40 | 31.39 | 48.53 | 27.19 | 51.11 | 6.04 | **26.67** | 35.62 | 75.00 |
| | SFT | 54.83 | 88.60 | 21.42 | 54.04 | 20.13 | 53.33 | 2.71 | 16.67 | 26.25 | 67.50 |
| | EAFT | 55.23 | **90.20** | 23.85 | 63.60 | 19.97 | 52.59 | 3.33 | 13.33 | 28.75 | 80.00 |
| | OverTone | 35.58 | 82.80 | 17.78 | 57.35 | 11.43 | 44.74 | 1.25 | 13.33 | 16.72 | 67.50 |
| | DFT | 70.92 | 86.00 | 32.42 | 47.79 | 35.07 | 58.22 | 8.75 | 16.67 | 45.78 | 75.00 |
| | TALR | 70.12 | 89.40 | **40.46** | 61.03 | 34.38 | 60.00 | 7.29 | **26.67** | 43.75 | 80.00 |
| | RANKTUNER | **72.38** | **90.20** | 38.26 | **65.44** | **36.25** | **64.00** | **10.21** | **26.67** | 46.56 | **85.00** |
| | $\Delta_{\text{Orig}}$ | ↑ 7.24 | ↑ 2.80 | ↑ 6.87 | ↑ 16.91 | ↑ 9.06 | ↑ 12.89 | ↑ 4.17 | ↑ 0.00 | ↑ 10.94 | ↑ 10.00 |
| | $\Delta_{\text{Best}}$ | ↑ 1.46 | ↑ 0.00 | ↓ 2.20 | ↑ 1.84 | ↑ 1.18 | ↑ 4.00 | ↑ 1.46 | ↑ 0.00 | ↑ 0.78 | ↑ 5.00 |

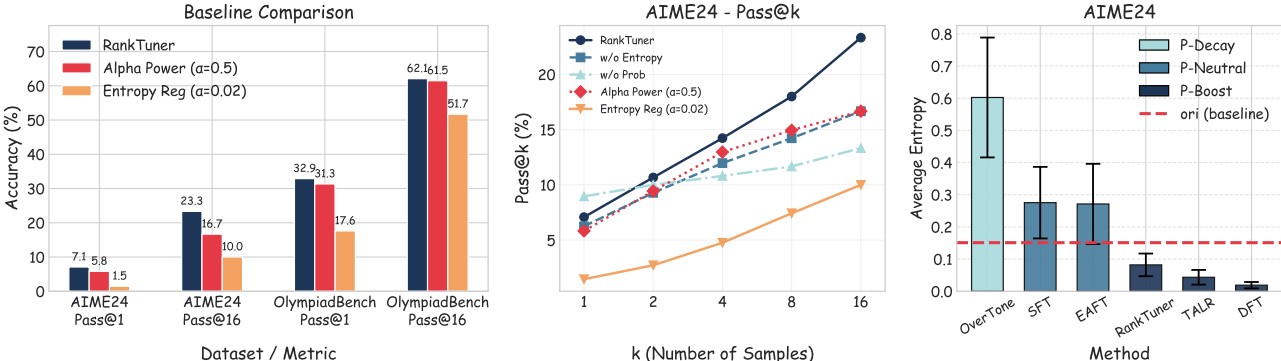

*Figure 4.* **Ablations, baselines, and inference entropy on AIME24 and OlympiadBench.** Left: We report Pass@1/Pass@16 and compare RANKTUNER with tuned *Alpha Power* ($\alpha$=0.5) and *Entropy Reg* ($\alpha$=0.02). Middle: We plot AIME24 Pass@k and further include two RANKTUNER ablations (w/o Prob, w/o Entropy), highlighting complementary roles of the probability- and entropy-aware terms. Right: We measure average inference entropy on AIME24 for Qwen2.5-Math-7B; the dashed line indicates the original (pre-finetuning) model and colors group methods by probability orientation (*P-decay*, *P-neutral*, *P-boost*).

*Table 3.* Out-of-distribution evaluation on ARC-C and GPQA using Qwen2.5-Math-7B. We report Pass@1 accuracy (higher is better). Best results are in bold and second-best results are underlined.

| Benchmark | Original | SFT | DFT | EAFT | TALR | RANKTUNER |
|---|---|---|---|---|---|---|
| ARC-C | 13.46 | 42.30 | 26.50 | 48.57 | 52.54 | **53.58** |
| GPQA | 7.86 | 25.00 | 27.90 | 25.63 | 29.29 | **29.64** |

and hurt generalization under distribution shift. Reweighted by relative rankings rather than a single signal, RANK-TUNER provides a richer and less distribution-specific training objective that preserves general reasoning ability.

## 5.4. RQ3: Key Ingredients of RANKTUNER

To isolate the effect of each ingredient, we conduct ablations on Qwen2.5-Math-7B. We compare against two strong loss-level baselines that reflect common alternatives to reweighting: **Alpha Power Loss** reshapes the token loss as $(1 - p^{\alpha})/\alpha$ (here $\alpha$=0.5), while **Entropy Regularization** augments CE with an entropy bonus $-\alpha H(p)$ (here $\alpha$=0.02) to encourage exploration/diversity.

Fig. 4 summarizes the results. The left and middle panels compare RANKTUNER with the two tuned baselines on AIME24 and OlympiadBench (Pass@1/Pass@16) and show AIME24 Pass@k, where RANKTUNER achieves consistent gains, especially at Pass@16. The middle panel further

includes two ablated variants: **RANKTUNER w/o Prob** (dropping the probability term $p_t^{-K(\xi_t)}$) can slightly improve Pass@1 but yields weaker improvements as $k$ grows, indicating reduced sample diversity/coverage; in contrast, **RANKTUNER w/o Entropy** (dropping the entropy term $H_t^{-K(\xi_t)}$) degrades across all $k$, showing the entropy component is essential for robust Pass@k gains. App. C.7 further compares dynamic $K(\xi_t)$ with fixed-exponent controls, confirming that uncertainty-aware exponent adaptation contributes beyond simply combining probability and entropy.

### 5.5. Inference-time Entropy Analysis

We study how *inference-time* token entropy changes after fine-tuning, and how it correlates with probability-oriented weighting designs. On **Qwen2.5-Math-7B**, we compute the average predictive entropy on AIME24 and then average over tokens. Specifically, we sample 8 decoding runs per prompt with temperature 0.2 and report entropy averaged across runs and tokens.

The right panel of Fig. 4 shows a striking pattern: the finetuned model's inference entropy is highly aligned with how weighting "steers" probability, forming three distinct signatures (*P-decay / P-neutral / P-boost*). OverTone (*P-decay*) yields the **highest** entropy—consistent with the idea that, in a *model-strong* reasoning setting, over-emphasizing currently-wrong tokens can amplify noisy supervision and make the model more "confused" (Li et al., 2025). In contrast, SFT/EAFT (*P-neutral*) exhibit a mild entropy rise; notably, EAFT being *entropy-weighted* does *not* automatically translate to lower *post*-finetuning inference entropy. Finally, *P-boost* methods reduce entropy, but with very different "sharpness": DFT shows the most aggressive entropy collapse. TALR uses a dynamic exponent, but it is still driven mainly via $p_t$ and does not explicitly account for token-type priors; whereas RANKTUNER stays closest to the original baseline by coupling the probability exponent to an uncertainty-linked term $K(\xi_t)$ tied to the rank-based proxy $\mathbb{E}[R_t]$.

### 6. Discussion

Our empirical evidence is currently strongest in the supervised fine-tuning setting. Broader RL-style post-training and multimodal fine-tuning tasks may also provide useful settings for applying probability–entropy calibration, but they are outside the core claims of this paper. More broadly, because RANKTUNER relies only on token-level predictive statistics, the same calibration principle could extend to LLM personalization fine-tuning (Liu et al., 2026; 2025a), multimodal/embodied adaptation (Ma et al., 2026), and LLM-based recommendation or generative information-extraction settings (Xu et al., 2024; Li et al., 2026). We leave a systematic study of these settings to future work.

### 7. Conclusion

We present RANKTUNER, a rank-guided token reweighting framework that calibrates downstream alignment by intrinsic uncertainty. By discretizing probability and entropy into a commensurate rank-based pair—the ground-truth rank and its expected rank—we derive a Relative Rank Indicator and use its inverse as a token-wise Relative Scale to focus updates on genuinely under-learned critical tokens, while down-weighting noisy or replaceable positions. Across multiple backbones and reasoning benchmarks, RANKTUNER achieves consistent gains over probability- or entropy-only reweighting baselines, and our ablations and entropy-based behavioral analysis highlight the complementary roles of both components in improving accuracy without collapsing diversity. These results suggest that probability–entropy calibration offers a simple and effective principle for adaptive fine-tuning, and this perspective is promising to generalize to broader tasks and training paradigms.

### Acknowledgements

The research presented in this paper was partially supported by the Research Grants Council of the Hong Kong Special Administrative Region, China (CUHK 2300246, RGC C1043-24G), (CUHK 14203425, RGC GRF 2151317), and CUHK 7010870.

### Impact Statement

This paper presents a token-level reweighting method for supervised fine-tuning, aiming to improve training stability and downstream reasoning performance by calibrating probability- and entropy-based signals. As a general optimization technique, our approach may help practitioners build more reliable and sample-efficient models for scientific and educational applications. At the same time, improved fine-tuning procedures can contribute to increased capabilities of language models (e.g., mathematical reasoning or code generation), which may be misused in downstream settings. We therefore recommend that any deployment follow established responsible-release practices (e.g., access control, monitoring, and usage policies) and comply with applicable laws and norms. Our work does not involve human subjects, and we conduct experiments using publicly available datasets and models.

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

# Appendix

This appendix provides supplementary materials to support the main text, including additional theoretical details, algorithmic analysis, and extended experiments. Below we summarize what each part aims to accomplish:

- **Theoretical Analysis** (A): proofs and derivations that connect rank-, probability-, and entropy-based quantities.
    - Rank–probability bound (A.1).
    - Expected rank–entropy bound (A.2).
    - Cauchy Mean Value Theorem derivation (A.3).
    - Coefficient analysis across different regimes (A.4).
    - Boundedness/tightness under surrogate substitution (A.5).

- **Pseudocode and Analysis** (B): implementation-oriented details, complexity analysis, and diagnostics/visualizations.
    - Pseudocode (B.1).
    - Time and memory complexity (B.2).
    - Noise sensitivity diagnostic (B.3).
    - Token-level visualization of difficulty and correctness (B.4).
    - Experimental validation of tightness of bounds and entropy-regime coverage (B.5).
    - Rationale of selections of initial weight for different tasks (B.6).
    - Potential RL-style extension (B.7).

- **Supplementary Experiments** (C): additional experimental details and results.
    - Dataset statistics (C.1).
    - Baselines Details (C.2).
    - Metrics (C.3).
    - Supplementary cross-architecture results for mathematical reasoning (C.4).
    - Sensitivity to monotone choices of $(f, g)$ (C.5).
    - Selections of the $\xi$ approximation (C.6).
    - Ablation studies of fixed $K$ versus dynamic $K(\xi_t)$ (C.7).
    - Code fine-tuning and evaluation (C.8).

## A. Theoretical Analysis

### A.1. Rank–Probability Bound

**Lemma A.1** (Rank–Probability Bound). *Let the probability distribution at position $t$ be sorted such that $p_{t,\hat{1}} \geq p_{t,\hat{2}} \geq \cdots$. For the ground-truth token with probability $p_t$ and rank $R_t$, we have $R_t \leq 1/p_t$ (Eq. (4)).*

**Proof.** Let the probabilities be sorted in non-increasing order as $p_{t,\hat{1}} \geq p_{t,\hat{2}} \geq \cdots$. Let the ground-truth token at position $t$ have probability $p_t$, and let $R_t$ denote its (1-indexed) rank in this sorted list (breaking ties arbitrarily). Then $p_{t,\hat{R}_t} = p_t$, and for every $i \leq R_t$ we have $p_{t,\hat{i}} \geq p_{t,\hat{R}_t} = p_t$. Therefore,

$$1 = \sum_{i \geq 1} p_{t,\hat{i}} \geq \sum_{i=1}^{R_t} p_{t,\hat{i}} \geq \sum_{i=1}^{R_t} p_t = R_t p_t, \tag{12}$$

which implies $R_t \leq 1/p_t$.

### A.2. Expected Rank–Entropy Bound

**Lemma A.2** (Expected Rank–Entropy Bound). *The expected rank $\mathbb{E}[R_t]$ satisfies Eq. (5).*

**Proof.** Let the probability distribution over the vocabulary at position $t$ be sorted in non-increasing order, $p_{t,\hat{1}} \geq p_{t,\hat{2}} \geq \cdots$, and recall that $p_{\max,t} \triangleq p_{t,\hat{1}}$. Define a random variable $R_t \in \{1, 2, \ldots\}$ whose distribution is given by this sorted list:

$$\Pr(R_t = i) \triangleq p_{t,\hat{i}}, \qquad i \geq 1. \tag{13}$$

Then $\mathbb{E}[R_t] = \sum_{i \geq 1} i\, p_{t,\hat{i}}$, and the Shannon entropy (in bits) is

$$H_t = -\sum_{i \geq 1} p_{t,\hat{i}} \log_2 p_{t,\hat{i}}. \tag{14}$$

**Case 1: $H_t \geq 2$.** Set $A \triangleq \mathbb{E}[R_t]$. Consider the set of (not necessarily monotone) distributions $\{p_i\}_{i \geq 1}$ on $\{1, 2, \ldots\}$ with mean constraint $\sum_{i \geq 1} i\, p_i = A$. It is a classical maximum-entropy result, due to Jaynes (Jaynes, 1957) and widely used in the guessing literature (Massey, 1994), that under a fixed mean (average-energy) constraint the unique entropy maximizer is the geometric (Boltzmann) distribution

$$p_i^{\text{geom}} = \frac{1}{A-1} \left(1 - \frac{1}{A}\right)^i, \qquad i \geq 1, \tag{15}$$

which indeed satisfies $\sum_{i \geq 1} p_i^{\text{geom}} = 1$ and $\sum_{i \geq 1} i\, p_i^{\text{geom}} = A$. Therefore, for any distribution with mean $A$ (in particular, our $\{p_{t,\hat{i}}\}$),

$$H_t \leq h(p^{\text{geom}}), \tag{16}$$

where $h(\cdot)$ denotes entropy in bits.

For the geometric distribution (15), a direct calculation gives

$$h(p^{\text{geom}}) = \log_2(A-1) + A \log_2\left(\frac{A}{A-1}\right). \tag{17}$$

The function $\phi(A) \triangleq A \log_2\left(\frac{A}{A-1}\right)$ is strictly decreasing for $A > 1$ and satisfies $\phi(2) = 2$ and $\lim_{A \to \infty} \phi(A) = \log_2(e) < 2$; hence for all $A \geq 2$,

$$A \log_2\left(\frac{A}{A-1}\right) \leq 2. \tag{18}$$

Moreover, $h(p^{\text{geom}}) \geq 2$ if and only if $A \geq 2$ (with equality at $A = 2$). Since we are in the regime $H_t \geq 2$ and $H_t \leq h(p^{\text{geom}})$ by (16), we must have $A \geq 2$, and thus (18) applies. Combining (16), (17), and (18), we obtain

$$H_t \leq \log_2(A-1) + 2. \tag{19}$$

Rearranging yields

$$A = \mathbb{E}[R_t] \geq \tfrac{1}{4} 2^{H_t} + 1, \tag{20}$$

which is exactly the first case in Eq. (5).

**Case 2: $H_t < 2$.** This regime follows from a simple decomposition on whether the guess is correct on the first try:

$$\begin{aligned}
\mathbb{E}[R_t] &= 1 \cdot p_{t,\hat{1}} + \sum_{i \geq 2} i\, p_{t,\hat{i}} \\
&\geq 1 \cdot p_{t,\hat{1}} + 2 \sum_{i \geq 2} p_{t,\hat{i}} \\
&= p_{\max,t} + 2(1 - p_{\max,t}) = 2 - p_{\max,t},
\end{aligned} \tag{21}$$

which matches the second case in Eq. (5).

**An entropy-only variant for the low-entropy regime.** The low-entropy regime ($H_t < 2$) in Eq. (5) can be expressed purely in terms of $H_t$ rather than $p_{\max,t}$. Define

$$h(p) \triangleq H_b(p) + (1-p)\log_2(|\mathcal{V}| - 1), \tag{22}$$

where $H_b(p) \triangleq -p\log_2 p - (1-p)\log_2(1-p)$ is the binary entropy. By Fano's inequality (see, e.g., (Cover, 1999)), among all distributions on $|\mathcal{V}|$ outcomes with maximal mass $p_{\max,t}$, the entropy is maximized by placing the remaining mass uniformly over the other $|\mathcal{V}| - 1$ outcomes. Therefore,

$$
\begin{aligned}
H_t &\leq -p_{\max,t}\log_2 p_{\max,t} - (1-p_{\max,t})\log_2\left(\frac{1-p_{\max,t}}{|\mathcal{V}| - 1}\right) \\
&= H_b(p_{\max,t}) + (1-p_{\max,t})\log_2(|\mathcal{V}| - 1) \\
&= h(p_{\max,t}), 
\end{aligned}
\tag{23}
$$

Since $h$ is strictly decreasing on $[1/|\mathcal{V}|, 1]$, this inequality yields an upper bound $p_{\max,t} \leq h^{-1}(H_t)$. Substituting this into the second case of Eq. (5), we conclude that even in the $H_t < 2$ regime, $\mathbb{E}[R_t]$ is bounded from below by a function of entropy alone:

$$\mathbb{E}[R_t] \geq 2 - h^{-1}(H_t). \tag{24}$$

### A.3. Cauchy Mean Value Theorem Derivation

In this section, we provide the detailed derivation of Eq. (6) from the main text, which connects the difference $f(R) - f(\mathbb{E}[R])$ to the logarithmic ratio $\log_2(R/\mathbb{E}[R])$ via the Cauchy Mean Value Theorem.

**Setup and theorem statement.** Let $u(x) \triangleq f(x) = \frac{1}{\log_2(x+1)}$ be the transformation function used in defining the Relative Rank Indicator, and let $v(x) \triangleq \log_2(x)$ be an auxiliary function. The Cauchy Mean Value Theorem states that if $u$ and $v$ are continuous on $[\mathbb{E}[R], R]$ (assuming $\mathbb{E}[R] < R$ without loss of generality) and differentiable on $(\mathbb{E}[R], R)$, then there exists a point $\xi \in (\mathbb{E}[R], R)$ such that

$$\frac{u(R) - u(\mathbb{E}[R])}{v(R) - v(\mathbb{E}[R])} = \frac{u'(\xi)}{v'(\xi)}. \tag{25}$$

**Computing the derivatives.** We compute the derivatives of $u(t)$ and $v(t)$ with respect to $t$:

$$
\begin{aligned}
u'(t) &= \frac{d}{dt}\left[\frac{1}{\log_2(t+1)}\right] \\
&= -\frac{1}{[\log_2(t+1)]^2} \cdot \frac{d}{dt}[\log_2(t+1)] \\
&= -\frac{1}{[\log_2(t+1)]^2} \cdot \frac{1}{(t+1)\ln 2},
\end{aligned}
\tag{26}
$$

and

$$v'(t) = \frac{d}{dt}[\log_2(t)] = \frac{1}{t\ln 2}. \tag{27}$$

**Forming the derivative ratio.** Taking the ratio of the derivatives at the point $\xi$, we obtain

$$\frac{u'(\xi)}{v'(\xi)} = \frac{-\frac{1}{(\xi+1)[\log_2(\xi+1)]^2 \ln 2}}{\frac{1}{\xi\ln 2}} = -\frac{\xi}{(\xi+1)[\log_2(\xi+1)]^2}. \tag{28}$$

Observe that the factor $\ln 2$ appearing in both the numerator and denominator cancels, which explains why the final expression is independent of the logarithm base.

**Obtaining the final relation.** Substituting this derivative ratio back into Eq. (25) and noting that $v(R) - v(\mathbb{E}[R]) = \log_2 R - \log_2 \mathbb{E}[R]$, we arrive at

$$u(R) - u(\mathbb{E}[R]) = -\frac{\xi}{(\xi + 1)[\log_2(\xi + 1)]^2} \cdot (\log_2 R - \log_2 \mathbb{E}[R]), \tag{29}$$

which, recalling that $u(x) = f(x)$, gives Eq. (6) with the positive coefficient $K(\xi) = \frac{\xi}{(\xi+1)[\log_2(\xi+1)]^2}$.

## A.4. Coefficient Analysis Across Different Regimes

In this section, we analyze the behavior of the positive coefficient $K(\xi) = \frac{\xi}{(\xi+1)[\log_2(\xi+1)]^2}$ across different values of $\xi$ to understand when the approximation $K(\xi) \approx 0.5$ is valid.

**The regime $\xi \approx 1$.** For typical reasoning tokens observed in Fig. 2, both $R$ and $\mathbb{E}[R]$ are small integers close to 1. In this case, the intermediate value $\xi$ guaranteed by the Cauchy Mean Value Theorem also lies near 1. Evaluating $K(\xi)$ at $\xi = 1$:

$$\begin{aligned}
K(1) &= \frac{1}{(1 + 1)[\log_2(1 + 1)]^2} \\
&= \frac{1}{2 \cdot [\log_2(2)]^2} \\
&= \frac{1}{2 \cdot 1^2} = 0.5.
\end{aligned} \tag{30}$$

This justifies the approximation used in the main text for low-rank tokens.

**General behavior for $\xi \in [1, 10]$.** As $\xi$ increases, the denominator $(\xi + 1)[\log_2(\xi + 1)]^2$ grows faster than the numerator $\xi$, causing $K(\xi)$ to decrease. For instance:

- At $\xi = 2$: $K(2) = \frac{2}{3 \cdot [\log_2(3)]^2} \approx \frac{2}{3 \cdot 1.585^2} \approx 0.265$

- At $\xi = 5$: $K(5) = \frac{5}{6 \cdot [\log_2(6)]^2} \approx \frac{5}{6 \cdot 2.585^2} \approx 0.125$

- At $\xi = 10$: $K(10) = \frac{10}{11 \cdot [\log_2(11)]^2} \approx \frac{10}{11 \cdot 3.459^2} \approx 0.076$

**Implications for the approximation.** The coefficient $K(\xi)$ exhibits monotone decay as $\xi$ increases. For the majority of chain-of-thought tokens in mathematical reasoning datasets (where $R, \mathbb{E}[R] \in [1, 5]$), the approximation $K(\xi) \in [0.2, 0.5]$ holds, with $0.5$ serving as a reasonable central estimate. For tokens with very high uncertainty (large $\mathbb{E}[R]$), the coefficient becomes smaller, which further dampens the influence of rank differences—consistent with our design goal of emphasizing confident predictions and de-emphasizing low-probability regimes.

In summary, the transformation $f(R) - f(\mathbb{E}[R])$ is approximately proportional to $\log_2(\mathbb{E}[R]/R)$ with a coefficient near $0.5$ for typical reasoning tokens, and this coefficient naturally decreases for high-uncertainty contexts, aligning with the principle of uncertainty-aware weighting.

## A.5. Boundedness/Tightness under surrogate substitution

**Boundedness/Tightness under surrogate substitution.** By the Cauchy mean value theorem (cf. App. A.3), the relative rank indicator admits the power-law form

$$\mathcal{I}_t = \left(\frac{\mathbb{E}[R_t]}{R_t}\right)^{K(\xi_t)}, \tag{31}$$

where $\xi_t$ lies between $R_t$ and $\mathbb{E}[R_t]$ and $K(\cdot)$ is a positive, slowly varying coefficient. For typical reasoning tokens where $R_t$ and $\mathbb{E}[R_t]$ are small, the intermediate value $\xi_t$ is also small; in the extreme case $\xi_t \approx 1$, App. A.4 gives $K(1) = 0.5$, motivating the convenient choice $K_0 \triangleq 0.5$. By contrast, for large $\xi$, $K(\xi) \to 0$ (App. A.4), making $\mathcal{I}_t = (\mathbb{E}[R_t]/R_t)^{K(\xi_t)} \approx 1$ and thus largely trivial; hence we primarily discuss the small-$\xi$ regime.

Our method substitutes the rank-based quantities in Eq. (31) using the two bridge bounds in Sec. 4.3: (i) $R_t \leq 1/p_t$ (Eq. (4)), and (ii) $\mathbb{E}[R_t] \geq s(H_t)$ (Eq. (5)). Under the approximation $K(\xi_t) \approx K_0$, this yields the surrogate indicator

$$\widehat{\mathcal{I}}_t \triangleq \left(p_t \, s(H_t)\right)^{K_0}, \tag{32}$$

which is the quantity used in Eq. (8).

**One-sided boundedness.** Since $R_t \leq 1/p_t$ implies $p_t \leq 1/R_t$ and $\mathbb{E}[R_t] \geq s(H_t)$ implies $1/\mathbb{E}[R_t] \leq 1/s(H_t)$, we have

$$\frac{\mathbb{E}[R_t]}{R_t} = \frac{1/R_t}{1/\mathbb{E}[R_t]} \geq \frac{p_t}{1/s(H_t)} = p_t \, s(H_t),$$

and therefore

$$\mathcal{I}_t \geq \widehat{\mathcal{I}}_t. \tag{33}$$

Thus, replacing $(1/R_t, \, 1/\mathbb{E}[R_t])$ by $(p_t, \, 1/s(H_t))$ produces a conservative (lower-bounding) surrogate of $\mathcal{I}_t$.

**Tightness via continuity and empirical gaps.** Define the two approximation gaps (evaluated empirically in App. B.5):

$$\Delta_t^{(p)} \triangleq \left| \frac{1}{R_t} - p_t \right|, \qquad \Delta_t^{(H)} \triangleq \left| \frac{1}{s(H_t)} - \frac{1}{\mathbb{E}[R_t]} \right|. \tag{34}$$

Consider the map $F(a, b) = (a/b)^{K_0}$ with $a > 0$ and $b > 0$. On any compact domain bounded away from zero, $F$ is Lipschitz continuous; hence the substitution $(a, b) = (1/R_t, \, 1/\mathbb{E}[R_t]) \mapsto (p_t, \, 1/s(H_t))$ induces a controlled change in $\mathcal{I}_t$ that scales linearly with $\Delta_t^{(p)}$ and $\Delta_t^{(H)}$ (up to a constant depending on the chosen domain). As a future direction, one can further tighten this bound by restricting rank computations to an effective support (e.g., top-$k$), which implicitly bounds $R_t$ and $\mathbb{E}[R_t]$ to a smaller range and can reduce computation from $O(|\mathcal{V}|)$ to $O(k)$ per token. Empirically, App. B.5 shows that both gaps are concentrated near zero on real model outputs, which supports the tightness of the surrogate substitution in Eq. (32).

# B. Pseudocode and Analysis

## B.1. Pseudocode

Algorithm 1 presents the pseudocode for RankTuner-guided supervised fine-tuning. The key distinction from standard SFT lies in the computation of token-wise scale $\mathcal{S}_t$ (Lines 4–10), which dynamically reweights each token based on its relative competence. Note that for simplification and training stability, we remove the $\frac{\xi_t}{(\xi_t + 1)}$ multiplier from the original formulation of $K(\xi_t)$.

## B.2. Time and Memory Complexity

**Time complexity.** RankTuner imposes no asymptotic overhead beyond standard supervised fine-tuning. All per-token computations are performed within a single forward pass and require $O(|\mathcal{V}|)$ operations per token, the same complexity as computing the cross-entropy loss. Crucially, these operations are fully vectorized and executed at the batch level via efficient broadcasting primitives, enabling parallelization across all tokens in a batch.

Tab. 4 summarizes the computational steps required to derive the key quantities $R_t$, $H_t$, and $p_{\max,t}$ from the model's output logits $\mathbf{z}_t$. For rank computation, we broadcast the scalar logit $z_{t,y_t}$ to match the shape of the full logit vector $\mathbf{z}_t$ and perform element-wise comparison $\mathbf{z}_t \geq z_{t,y_t}$ in $O(|\mathcal{V}|)$ time, yielding a binary mask whose sum gives $R_t$. Entropy $H_t$ is computed via standard summation over the probability distribution $\mathbf{p}_t = \mathrm{softmax}(\mathbf{z}_t)$, and $p_{\max,t}$ is obtained via a reduction operation (e.g., max), both requiring $O(|\mathcal{V}|)$ time. The subsequent computation of $s(H_t)$, $K(\xi_t)$, and $\mathcal{S}_t$ involves only scalar arithmetic and is negligible ($O(1)$ per token).

**Memory complexity.** The memory footprint of RankTuner is identical to that of standard SFT. The logit tensor $\mathbf{z}_t$ and probability distribution $\mathbf{p}_t$ are already materialized during the forward pass for loss computation. Our method introduces only a handful of scalar variables per token ($R_t$, $H_t$, $p_{\max,t}$, $\mathcal{S}_t$), incurring $O(1)$ additional space per position. Across a batch of $B$ sequences with average length $T$, the total overhead is $O(BT)$, which is negligible compared to the $O(BT|\mathcal{V}|)$ memory required for storing logits.

---

**Algorithm 1** RankTuner-Guided Supervised Fine-Tuning

---

**Require:** Model $\mathcal{M}_\theta$, dataset $\mathcal{D}$, original token weights $\{w_t\}$
1: **for** each batch $\mathbf{x}, \mathbf{y}$ from $\mathcal{D}$ **do**
2:     $\mathbf{z} \leftarrow \mathcal{M}_\theta(\mathbf{x})$
3:     **for** each token position $t$ **do**
4:         $p_t \leftarrow p_\theta(y_t \mid \mathbf{x}_{<t})$      *Relative Scale Computing*
5:         $R_t \leftarrow \text{Rank}(z_{t,y_t}; \mathbf{z}_t)$
6:         $H_t \leftarrow -\sum_i p_{t,i} \log_2 p_{t,i}$
7:         $s(H_t) \leftarrow \begin{cases} \frac{1}{4} \cdot 2^{H_t} + 1, & H_t \geq 2 \\ 1 + (1 - p_{\max,t}), & H_t < 2 \end{cases}$
8:         $\xi_t \leftarrow \max\big(R_t, s(H_t)\big)$
9:         $K(\xi_t) \leftarrow [\log_2(\xi_t + 1)]^{-2}$
10:       $\mathcal{S}_t \leftarrow \big(p_t \cdot s(H_t)\big)^{-K(\xi_t)}$
11:       $\widetilde{w}_t \leftarrow w_t \cdot \mathcal{S}_t$
12:     **end for**
13:     $\mathcal{L} \leftarrow \frac{1}{T} \sum_{t=1}^{T} \widetilde{w}_t \cdot \ell_t$
14:     Update $\theta$ via gradient descent on $\mathcal{L}$
15: **end for**

---

*Table 4.* Computational breakdown of key quantities in RankTuner. All operations are vectorized at the batch level and incur $O(|\mathcal{V}|)$ complexity per token.

| Quantity | Operation | Complexity |
|:---:|:---|:---:|
| $R_t$ | Broadcast $z_{t,y_t}$, compare $\mathbf{z}_t \geq z_{t,y_t}$, sum | $O(|\mathcal{V}|)$ |
| $H_t$ | Compute $-\sum_v p_{t,v} \log_2 p_{t,v}$ over $\mathbf{p}_t$ | $O(|\mathcal{V}|)$ |
| $p_{\max,t}$ | Reduction $\max(\mathbf{p}_t)$ | $O(|\mathcal{V}|)$ |
| $s(H_t), K(\xi_t), \mathcal{S}_t$ | Scalar arithmetic on $R_t, H_t, p_{\max,t}$ | $O(1)$ |

## B.3. Noise Sensitivity Diagnostic

We stress-test whether a token-importance signal is *noise-attractive* (i.e., prone to assigning high scores to irrelevant tokens) via a controlled *noise insertion* procedure on a clean instruction-following dataset, and then measure how strongly different indicators "surface" the injected noise.

**Datasets.** We take a subset of $N=1000$ instruction–response pairs from **NuminaMath-CoT** (Jia et al., 2024), formatted in an Alpaca-style schema with fields `instruction`, `input`, and `output`. As a source of semantically irrelevant text, we use the **Stanford Alpaca** instruction-following data (Taori et al., 2023) and extract noise sentences from its `output` fields.

**Noise construction.** We set the corruption ratio to $\rho = 0.1$ and corrupt 10% of examples by inserting a semantically irrelevant sentence. Concretely, for each selected NuminaMath-CoT example, we keep its prompt unchanged (concatenating `instruction` and `input` when present), sample a random Alpaca example, and take the *first sentence* from its `output` as noise $\eta_i$. We then insert $\eta_i$ into the *middle* of the reference response $y_i$ at the nearest whitespace around the midpoint:

$$y_i^{\text{noisy}} = y_i^{\text{pre}} \parallel \eta_i \parallel y_i^{\text{post}}.$$

**Token-level indicators.** For each response token position $t$ (i.e., positions after the prompt) of example $i$, we compute three scores:

$$s_{i,t}^{\text{ent}} = H_{i,t}, \qquad s_{i,t}^{\text{prob}} = -\log(p_{i,t}), \qquad s_{i,t}^{\text{ours}} = \frac{1}{\mathcal{I}_{i,t}},$$

where $p_{i,t}$ is the ground-truth probability, $H_{i,t}$ is the predictive entropy, and $\mathcal{I}_{i,t}$ is our relative-rank indicator (higher $s$ means "more important/harder").

**Token-level noise precision/recall.** Let $\mathcal{T}$ be the set of all response-token indices across all examples (after tokenization and truncation), and let $\mathcal{N} = \bigcup_{i \in \mathcal{C}} \mathcal{N}_i$ be the set of all injected noise tokens across corrupted examples. For a method $m \in \{\text{ent}, \text{prob}, \text{ours}\}$, we rank all tokens in $\mathcal{T}$ by $s_{i,t}^m$ in descending order and take the top fraction $\rho$:

$$K = \lceil \rho |\mathcal{T}| \rceil, \quad \mathcal{T}_{\text{top}}^m = \text{Top-}K\big(\{(i,t) \in \mathcal{T}\}, s_{i,t}^m\big).$$

We then report

$$\text{Prec}^m = \frac{|\mathcal{T}_{\text{top}}^m \cap \mathcal{N}|}{|\mathcal{T}_{\text{top}}^m|}, \qquad \text{Rec}^m = \frac{|\mathcal{T}_{\text{top}}^m \cap \mathcal{N}|}{|\mathcal{N}|}.$$

**Sequence-level (span) scoring and noise hit.** For each example $i$, we define a span $\mathcal{S}_i$ of length $L_i$ in token space. If $i \in \mathcal{C}$, we set $\mathcal{S}_i = \mathcal{N}_i$ (the injected noise span). If $i \notin \mathcal{C}$, we select a *length-matched mid-span* inside the response:

$$\mathcal{S}_i = \{t_0, t_0+1, \ldots, t_0+L_i-1\}, \quad t_0 = \text{prompt\_len}_i + \left\lfloor \frac{\text{out\_len}_i - L_i}{2} \right\rfloor,$$

where prompt_len and out_len are tokenized lengths (after truncation) of the prompt and response, respectively. We aggregate span scores by averaging:

$$S_i^m = \frac{1}{|\mathcal{S}_i|} \sum_{t \in \mathcal{S}_i} s_{i,t}^m.$$

We rank examples by $S_i^m$ in descending order, take the top $\lceil \rho N \rceil$ examples, and report the *noise hit*:

$$\text{Hit}_{\text{seq}}^m = \sum_{i \in \text{Top-}\lceil \rho N \rceil(\{1,\ldots,N\}, S_i^m)} \mathbb{I}[i \in \mathcal{C}].$$

Lower $\text{Hit}_{\text{seq}}^m$ indicates less tendency to surface the injected noise as "important" at the sequence level.

**Illustrative example.** Below is a simplified excerpt of one corrupted sample:

| Corrupted sample (simplified excerpt) | |
|---|---|
| **Prompt (NuminaMath).** | Given the functions $f(x) = \log_a(1+x)$ and $g(x) = \log_a(1-x)$, where $a > 0$ and $a \neq 1, \ldots$ |
| **Noise sentence (Alpaca).** | Aerobic and anaerobic exercise are two types of exercises that work differently on the body. |
| **Noisy response (excerpt).** | ... therefore, $f(x) - g(x)$ is an odd function. *[noise inserted here]* From $f(x) - g(x) > 0$, we get ... |

## B.4. Token-Level Visualization of Difficulty and Correctness

Fig. 5 shows that the key trend is unchanged: relatively replaceable or noisy tokens are down-weighted, while genuinely incorrect result-critical tokens receive stronger emphasis. This illustrates how entropy helps separate intrinsic ambiguity from true errors when combined with the ground-truth probability signal.

## B.5. Experimental Validation of Tightness of Bounds

We empirically validate the tightness of the two key bounds used throughout the paper (Sec. 4.3) by measuring their approximation errors on chain-of-thought tokens from Minerva Math predicted by Qwen3-8B. Specifically, we examine: (i) the rank–probability gap $\frac{1}{R_t} - p_t \in [0,1)$, which probes how well $1/R_t$ serves as a discrete surrogate for the ground-truth probability $p_t$; and (ii) the inverse expected-rank gap $\frac{1}{s(H_t)} - \frac{1}{\mathbb{E}[R_t]} \in [0,1)$, where $s(H_t)$ is the entropy-based lower bound on $\mathbb{E}[R_t]$ defined in Eq. (5). Fig. 6, Fig. 3, and Tab. 6 show that both approximation gaps are concentrated near zero (computed over 4k+ tokens): the median errors are $0.0259$ for $1/R - p$ and $0.0093$ for $1/s(H) - 1/\mathbb{E}[R]$, and even at the 90th percentile the errors remain moderate ($\leq 0.348$ and $\leq 0.297$, respectively). This supports our use of rank-based surrogates: $1/R_t$ is a practical proxy for $p_t$ (consistent with the envelope $R \leq 1/p$), and $1/\mathbb{E}[R_t]$ closely tracks its entropy-induced theoretical bound $1/s(H_t)$, making either quantity a reliable stand-in for the other when constructing uncertainty-aware competence and scaling signals.

*Table 5.* Token statistics for Qwen3-8B on five mathematical reasoning benchmarks under the entropy split $H_t = 2$. We report the full dataset size, the loaded subset used for this diagnostic, and token counts and shares on each side of the threshold. For datasets with chain-of-thought solutions, both CoT tokens and final-answer tokens are counted; when no solution field is available and evaluation falls back to answer-only supervision, only answer tokens are counted.

| Dataset | Dataset Size | Loaded | Total Tokens | $H_t < 2$ Tokens | $H_t \geq 2$ Tokens | $H_t < 2$ Ratio | $H_t \geq 2$ Ratio |
|---|---|---|---|---|---|---|---|
| AIME24 | 30 | 30 | 49,069 | 45,363 | 3,706 | 92.45% | 7.55% |
| MATH-OAI | 500 | 50 | 10,053 | 9,891 | 162 | 98.39% | 1.61% |
| Minerva Math | 272 | 50 | 5,904 | 5,646 | 258 | 95.63% | 4.37% |
| OlympiadBench | 675 | 50 | 72,674 | 68,006 | 4,668 | 93.58% | 6.42% |
| AMC23 | 40 | 40 | 151 | 81 | 70 | 53.64% | 46.36% |

*Table 6.* Summary statistics of approximation errors (smaller is better). We report robust central tendency and moderate quantiles to highlight that the errors are typically small.

| ERROR TYPE | MEAN | MEDIAN | STD | P80 | P90 |
|---|---|---|---|---|---|
| $1/R - p$ | 0.109776 | 0.025879 | 0.151621 | 0.228027 | 0.348145 |
| $1/s(H) - 1/\mathbb{E}[R]$ | 0.084548 | 0.009272 | 0.137787 | 0.167955 | 0.297379 |

Fig. 2 is a local visualization of the bound, not a global entropy histogram. To check whether the token mass is confined to a single entropy regime, Tab. 5 reports full token statistics for Qwen3-8B across five mathematical reasoning benchmarks using the split $H_t = 2$. The mass is not exclusively concentrated below the threshold: AIME24 and OlympiadBench still contain non-trivial $H_t \geq 2$ fractions, and AMC23 has a sizable $H_t \geq 2$ share because this diagnostic uses answer-only tokens when no solution field is available. This matters for weighting design because even minority high-entropy tokens can exert disproportionate training impact (Wang et al., 2025); therefore, entropy-aware calibration remains motivated rather than reducible to only modeling the majority low-entropy regime.

### B.6. Rationale of Selections of Initial Weight for Different Tasks

For all fine-tuning tasks on math reasoning datasets, we set $w_t = p_t$ as the initial weight, which corresponds to the ground-truth probability of the token. For general fine-tuning tasks, we set $w_t = 1$ as the initial weight, which represents a uniform weighting scheme. We provide the rationale for these selections from three perspectives.

1. **A knowledge–noise separation view explains why we initialize $w_t = p_t$ for math reasoning but $w_t = 1$ for general tasks.** For math reasoning datasets, most of the knowledge space lies in the high $p_t$ region, indicating that the model is already well-aligned with the pretraining math datasets. As illustrated in Fig. 7, setting $w_t = p_t$ helps distinguish the knowledge region from the noise region and reduces the contribution of noise. In contrast, for most common tasks, the majority of the knowledge space resides in the low $p_t$ region. Therefore, setting $w_t = 1$ preserves the basic trend of the NLL loss, and the gradient will be more assigned to the low $p_t$ region.

2. **An importance-sampling view of SFT suggests $w_t = p_t$ is a variance-stable starting point, and composes naturally with our scale.** Standard SFT takes gradients under a fixed demonstration distribution. Following (Wu et al., 2026), we can rewrite the SFT gradient as an on-policy expectation under the model distribution by inserting the importance ratio between the Dirac-delta action distribution and the model policy:

$$\mathbb{E}_{(x,y)\sim\mathcal{D}}[-\nabla_\theta \log \pi_\theta(y_t \mid y_{<t}, x)] = \mathbb{E}_{x\sim\mathcal{D}_x} \mathbb{E}_{\hat{y}_t\sim\pi_\theta(\cdot|y_{<t},x)} \left[ \frac{\mathbb{I}(\hat{y}_t = y_t)}{\pi_\theta(\hat{y}_t \mid y_{<t}, x)} \left( -\nabla_\theta \log \pi_\theta(\hat{y}_t \mid y_{<t}, x) \right) \right]. \quad (35)$$

The importance weight above is $\frac{1}{\pi_\theta(\hat{y}_t|y_{<t},x)}$, which becomes $\frac{1}{p_t}$ on the (only) contributing event $\hat{y}_t = y_t$. This highlights a simple stability consideration: multiplying by $p_t$ *neutralizes* the potentially large $\frac{1}{p_t}$ factor at the ground-truth action, yielding a unit effective weight and reducing variance, while keeping the same update direction toward increasing $\pi_\theta(y_t \mid y_{<t}, x)$. Under our unified weighted-NLL view, this corresponds to choosing $w_t = p_t$, after which our RankTuner scale $\mathcal{S}_t$ (Eq. (10)) can be introduced as an additional multiplicative correction in a standard importance-weight form.

3. **A logit-gradient view links our weighting choice to an adaptive loss shape that interpolates across downstream regimes.** Following the logit-gradient perspective in (Li et al., 2025), Fig. 7 compares the normalized logit-gradient

magnitude $W_f(p) = -f'(p)\,p(1-p)$ induced by three representative loss shapes: $f(p) = -\log p$ (standard SFT), $f(p) = -p$ (DFT), and $f(p) = (1 - p^{0.5})/0.5$, which becomes close to RankTuner when approximating $K(\xi) \approx K(1) = 0.5$ and using $w_t = p_t$. Under this view, RankTuner behaves like an adaptive power loss with exponent $1 - K(\xi)$ together with an entropy-dependent scaling factor $s(H)^{-K(\xi)}$, enabling it to smoothly interpolate across downstream regimes from model-strong to model-weak settings.

## B.7. Potential RL-style Extension

RankTuner can also be viewed as a token-level correction that could be composed with RL-style post-training, although we do not evaluate this setting in the present work. For a PPO/GRPO-style update with token-level policy ratio

$$\eta_t(\theta) = \frac{\pi_\theta(y_t \mid y_{<t}, x)}{\pi_{\theta_{\text{old}}}(y_t \mid y_{<t}, x)}, \tag{36}$$

one possible extension is to modulate the ratio by the Relative Scale:

$$\widetilde{\eta}_t(\theta) = \eta_t(\theta) \cdot \mathcal{S}_t. \tag{37}$$

This form would emphasize token positions where the model underperforms relative to the entropy-induced expectation, while leaving reward design and advantage estimation unchanged. We regard this only as a future direction, since our experiments focus on supervised fine-tuning objectives and do not validate RL variants.

# C. Supplementary Experiments

## C.1. Datasets Statistics

Tab. 7 summarizes the all datasets used in this paper to assess both in-domain effectiveness and out-of-domain generalization. We fine-tune models on two complementary training corpora: NuminaMath-CoT-10k targets mathematical reasoning with explicit chain-of-thought supervision, while Evol-Instruct-Code-80k focuses on code synthesis and execution-oriented problem solving. Evaluation is conducted along three axes. First, in-domain mathematical benchmarks (AIME24, AMC23, MATH-OAI, Minerva Math, OlympiadBench) measure improvements in rigorous multi-step reasoning after math-centric training. Second, out-of-distribution test sets (ARC-C, GPQA) probe whether the gains transfer beyond the training distribution to broader scientific and knowledge-intensive reasoning, reflecting robustness and generalization. Third, code generation benchmarks (HumanEval, HumanEval+) quantify functional coding ability and help verify that performance gains do not come at the expense of programming competence. Together, this diversified suite provides a comprehensive basis for demonstrating the effectiveness and generalizability of our method.

## C.2. Baselines Details

To evaluate the effectiveness of **RankTuner**, we compare it against several representative fine-tuning methods. For consistency, all methods are formulated within a unified weighting framework where the objective is to minimize the weighted negative log-likelihood (NLL) loss:

$$\mathcal{L}(\theta) = \mathbb{E}_{(x,y)\sim\mathcal{D}} \left[ -\sum_{t=1}^{T} w_t \log p_t \right], \tag{38}$$

where $p_t = \pi_\theta(y_t \mid y_{<t}, x)$ is the probability of the ground-truth token $y_t$ at decoding step $t$. The weighting coefficient $w_t$ for each baseline is defined as follows:

- **SFT**: Standard Supervised Fine-Tuning treats every token as equally important, assigning a uniform weight $w_t = 1$. This approach serves as the primary baseline but is prone to overfitting on easy tokens and catastrophic forgetting of general capabilities.

- **OverTone** (Liu et al., 2025b): OverTone employs token-level smoothing with a *skip* mechanism: it mixes the ground-truth label with the model's filtered prediction only when the mixed target still places the highest probability on the ground-truth token; otherwise it skips mixing and falls back to the one-hot label. In our experiments, we use OverTone

*Table 7.* Overview of datasets used in this study. Training sets are used for model fine-tuning, while test sets evaluate mathematical reasoning and code generation capabilities. OOD test sets assess model generalization to out-of-distribution scenarios when fine-tuned on the mathematical training datasets. For readability, section pointers are shown once per dataset group in the group header row (right-aligned) using the prefixes *Sec.* (Section).

| Dataset | Type | Size | Source | Reference |
|---------|------|------|--------|-----------|
| ***Mathematical Reasoning*** | | | | *Sec. 5.2; C.4* |
| NuminaMath-CoT-10k | Train | 10K | HuggingFace | (Jia et al., 2024) |
| AIME24 | Test | 30 | HuggingFace | AIME 2024 |
| AMC23 | Test | 40 | HuggingFace | AMC 2023 |
| MATH-OAI | Test | 500 | HuggingFace | (Lightman et al., 2024) |
| Minerva Math | Test | 272 | HuggingFace | (Lewkowycz et al., 2022) |
| OlympiadBench | Test | 8,476 | GitHub | (He et al., 2024) |
| ***Out-of-Distribution Test Sets*** | | | | *Sec. 5.3* |
| ARC-C | OOD Test | 2,590 | HuggingFace | (Clark et al., 2018) |
| GPQA | OOD Test | 448 | HuggingFace | (Rein et al., 2024) |
| ***Code Generation*** | | | | *Sec. C.8* |
| Evol-Instruct-Code-80k | Train | 78,264 | HuggingFace | (Luo et al., 2023) |
| HumanEval | Test | 164 | HuggingFace | (Chen et al., 2021) |
| HumanEval+ | Test | 164 | GitHub | (Liu et al., 2023) |

with hyperparameters of their LoRA implementation. *For presentation under our unified weighted-NLL framework* (not the exact baseline implementation), this behavior can be approximated as a skip-gated discrete reweighting: $w_t = 1 - (1 - \lambda)\mathbb{I}(p_t = p_{\max})$ (typically $\lambda = 0.1$), where $p_t = \pi_\theta(y_t \mid y_{<t}, x)$ and $p_{\max} = \max_v \pi_\theta(v \mid y_{<t}, x)$.

- **DFT** (Wu et al., 2026): Dynamic Fine-Tuning rescales the loss using the stop-gradient of the target token probability, $w_t = \text{sg}(p_t)$. By prioritizing tokens where the model is already relatively confident, DFT stabilizes gradient updates and improves generalization from a reinforcement learning perspective.

- **EAFT** (Diao et al., 2026): Entropy-Adaptive Fine-Tuning utilizes normalized Top-$K$ token entropy $H_t^{\text{top-}K}$ as a gating mechanism, $w_t = \tilde{H}_t = H_t^{\text{top-}K}/\ln K$, where $K$ is the number of top tokens used for entropy approximation. In our implementation, we set $K = 20$ and approximate $\ln K \approx 3$ for computational efficiency following the original implementation. This method suppresses gradients on "Confident Conflict" tokens to preserve the model's general capabilities.

- **TALR** (Lin et al., 2026): Token-Adaptive Loss Reweighting down-weights "hard" tokens by exponentially tilting the token loss: $w_t \propto \exp(-\ell_t/\tau)$, where $\ell_t = -\log p_t$ is the token-level NLL. This can be simplified as $w_t \propto p_t^{1/\tau}$. The temperature $\tau$ is set dynamically as the median of the per-sequence average loss within the current training batch, serving as a scale that controls the sharpness of reweighting. In practice, TALR uses stop-gradient on the weight and applies a floor to avoid vanishing contributions, e.g., $w_t = \max(\text{sg}(p_t^{1/\tau}), w_{\min})$ with $w_{\min} = 0.01$.

The following table summarizes the weighting mechanisms of the baselines. Note that for some methods (e.g., OverTone and TALR), the formulas shown are approximations from a weighting perspective under our unified framework, rather than their exact original implementations:

### C.3. Metrics

We evaluate model performance using the *Pass@k* metric, which measures the probability that at least one correct solution is found among $k$ sampled attempts. For each problem, we generate $n = 16$ independent solution samples with temperature 1.0 and top-$p$ 1.0. To compute Pass@k for $k \in \{1, 2, 4, 8, 16\}$, we employ a combinatorial approach that considers all possible combinations of $k$ samples from the $n$ generated samples.

Formally, for a given problem with $n$ samples, let $\mathcal{S} = \{s_1, s_2, \ldots, s_n\}$ denote the set of samples, where each sample $s_i$ has

*Table 8.* Summary of Baseline Weighting Mechanisms

| METHOD | WEIGHTING FORMULA ($w_t$) | CORE SIGNAL |
|---|---|---|
| SFT | 1 | UNIFORM |
| OVERTONE (LIU ET AL., 2025B) | $\approx 1 - (1-\lambda)\mathbb{I}(p_t = p_{max})$ | GT PROBABILITY (GATED) |
| DFT (WU ET AL., 2026) | $p_t$ | GT PROBABILITY |
| EAFT (DIAO ET AL., 2026) | $H_t / \log K$ | TOKEN ENTROPY |
| TALR (LIN ET AL., 2026) | $\approx p_t^{1/\tau}$ | GT PROBABILITY |

*Table 9.* Performance comparison on mathematical reasoning benchmarks for additional model architectures. We report Pass@1 and Pass@16 metrics. Best results for each base model are in bold. $\Delta_{\mathrm{Orig}}$ denotes RANKTUNER minus the Original base model, while $\Delta_{\mathrm{Best}}$ denotes RANKTUNER minus the best non-RANKTUNER fine-tuning baseline in the same block.

| Model | Method | MATH-OAI | | Minerva Math | | OlympiadBench | | AIME24 | | AMC23 | |
|---|---|---|---|---|---|---|---|---|---|---|---|
| | | P@1 | P@16 | P@1 | P@16 | P@1 | P@16 | P@1 | P@16 | P@1 | P@16 |
| Qwen2.5-Math-1.5B | Original | 23.11 | 82.20 | 5.79 | 34.93 | 13.82 | 52.74 | 2.29 | **23.33** | 17.97 | **75.00** |
| | SFT | 43.91 | 82.60 | 11.74 | 42.28 | 14.10 | 47.85 | 0.42 | 3.33 | 17.50 | 57.50 |
| | EAFT | 42.90 | 82.60 | 12.02 | 43.01 | 12.86 | 45.04 | 0.42 | 3.33 | 17.66 | 65.00 |
| | DFT | 62.39 | 82.60 | 21.21 | 41.91 | 26.82 | 52.44 | 5.21 | 16.67 | 34.53 | 72.50 |
| | TALR | **62.94** | 86.80 | **26.42** | **53.68** | **27.34** | 53.19 | **6.46** | 20.00 | **35.00** | 70.00 |
| | RANKTUNER | 62.00 | **87.80** | 23.30 | 51.47 | 26.57 | **56.74** | 5.83 | 20.00 | 33.59 | 70.00 |
| | $\Delta_{\mathrm{Orig}}$ | ↑ 38.89 | ↑ 5.60 | ↑ 17.51 | ↑ 16.54 | ↑ 12.75 | ↑ 4.00 | ↑ 3.54 | ↓ 3.33 | ↑ 15.63 | ↓ 5.00 |
| | $\Delta_{\mathrm{Best}}$ | ↓ 0.94 | ↑ 1.00 | ↓ 3.12 | ↓ 2.21 | ↓ 0.77 | ↑ 3.55 | ↓ 0.63 | ↑ 0.00 | ↓ 1.41 | ↓ 2.50 |
| Qwen3-4B | Original | 68.58 | **89.60** | 32.88 | 50.37 | 30.23 | 54.07 | 10.00 | **26.67** | 41.88 | 70.00 |
| | SFT | 51.34 | 88.00 | 16.89 | 50.00 | 18.42 | 53.19 | 3.33 | 20.00 | 23.44 | 75.00 |
| | EAFT | 49.08 | 88.00 | 18.59 | 58.46 | 16.60 | 48.74 | 3.33 | 16.67 | 24.69 | 77.50 |
| | DFT | 66.09 | 84.40 | 29.89 | 43.38 | 31.53 | 53.19 | 6.88 | 13.33 | 37.50 | 70.00 |
| | TALR | 67.24 | 88.20 | **33.71** | 55.15 | 30.83 | 57.78 | 6.46 | 16.67 | 40.47 | 80.00 |
| | RANKTUNER | **67.35** | 89.60 | 33.50 | **61.76** | **32.71** | **60.89** | **9.58** | **26.67** | 41.09 | **82.50** |
| | $\Delta_{\mathrm{Orig}}$ | ↓ 1.23 | ↑ 0.00 | ↑ 0.62 | ↑ 11.40 | ↑ 2.48 | ↑ 6.81 | ↓ 0.42 | ↑ 0.00 | ↓ 0.78 | ↑ 12.50 |
| | $\Delta_{\mathrm{Best}}$ | ↑ 0.11 | ↑ 1.40 | ↓ 0.21 | ↑ 3.30 | ↑ 1.18 | ↑ 3.11 | ↑ 2.70 | ↑ 6.67 | ↑ 0.62 | ↑ 2.50 |
| Llama-3.1-8B | Original | 1.74 | 15.80 | 1.24 | 12.87 | 0.91 | 10.07 | 0.00 | 0.00 | 1.56 | 17.50 |
| | SFT | 17.18 | 60.40 | 4.96 | 29.04 | 3.49 | 24.44 | 0.42 | 3.33 | 5.16 | 47.50 |
| | EAFT | 15.94 | 59.60 | 5.06 | 29.41 | 3.50 | 25.93 | 0.00 | 0.00 | 5.78 | 40.00 |
| | DFT | 26.24 | 58.60 | 7.24 | 27.57 | 6.82 | 26.81 | 0.63 | 6.67 | 12.34 | 35.00 |
| | TALR | 27.03 | 63.60 | 7.70 | 34.93 | 6.73 | 30.96 | 0.21 | 3.33 | 9.06 | 42.50 |
| | RANKTUNER | **28.66** | **67.00** | **9.26** | **37.13** | **7.99** | **34.07** | **0.83** | **6.67** | **12.66** | **50.00** |
| | $\Delta_{\mathrm{Orig}}$ | ↑ 26.93 | ↑ 51.20 | ↑ 8.02 | ↑ 24.26 | ↑ 7.08 | ↑ 24.00 | ↑ 0.83 | ↑ 6.67 | ↑ 11.09 | ↑ 32.50 |
| | $\Delta_{\mathrm{Best}}$ | ↑ 1.63 | ↑ 3.40 | ↑ 1.56 | ↑ 2.20 | ↑ 1.17 | ↑ 3.11 | ↑ 0.20 | ↑ 0.00 | ↑ 0.32 | ↑ 2.50 |

a binary correctness score $c_i \in \{0, 1\}$. For each value of $k$, we enumerate all $\binom{n}{k}$ combinations of $k$ samples. A combination $\mathcal{C} \subseteq \mathcal{S}$ with $|\mathcal{C}| = k$ is considered to *pass* if at least one sample in $\mathcal{C}$ is correct, i.e., $\max_{s_i \in \mathcal{C}} c_i = 1$. The Pass@$k$ metric is then computed as:

$$\text{Pass@}k = \frac{\sum_{\text{problem } p} \sum_{\mathcal{C} \in \binom{s_p}{k}} \mathbb{I}\left[\max_{s_i \in \mathcal{C}} c_i = 1\right]}{\sum_{\text{problem } p} \binom{|\mathcal{S}_p|}{k}} \times 100\%, \tag{39}$$

where $\mathcal{S}_p$ denotes the set of samples for problem $p$, and $\mathbb{I}[\cdot]$ is the indicator function. Intuitively, Pass@1 is simply the *expected one-shot accuracy*: the probability that a single independent sample solves the problem. In contrast, Pass@16 measures the probability that *at least one* of the $n{=}16$ independent samples succeeds, and is therefore more sensitive to whether the sampler can cover diverse reasoning paths (i.e., solution diversity/coverage) rather than only improving the most likely trajectory.

## C.4. Supplementary Cross-Architecture Results for Mathematical Reasoning

Tab. 9 shows that RANKTUNER consistently improves mathematical reasoning across architectures, from Qwen2.5-Math-1.5B (Yang et al., 2024) and Qwen3-4B (Yang et al., 2025) to Llama-3.1-8B (Grattafiori et al., 2024), while also reporting direct gaps to the strongest competing fine-tuning baseline. The gains often concentrate on Pass@16 (e.g., Qwen3-4B on Minerva Math/OlympiadBench and AMC23), suggesting better coverage of diverse reasoning paths. We also observe a

*Table 10.* Pass@1 (%) on Qwen2.5-Math-7B under alternative monotone choices of $f$ and $g$. The first row is a formula-instantiated approximation that directly applies the figure expression with Eqs. (4) and (5) to approximate $R$ and $\mathbb{E}[R]$, so it is slightly different from the final RANKTUNER configuration. Best values in each benchmark column are in bold.

| $f(x)$ | $g(z)$ | AIME24 | AMC23 | MATH-OAI | Minerva Math | OlympiadBench |
|---|---|---|---|---|---|---|
| $\frac{1}{\log_2(1+x)}$ | $2^z$ | 7.50 | **45.31** | **69.54** | 27.53 | **33.91** |
| $\frac{1}{\log_2(1+x)}$ | $e^{0.5z}$ | 8.12 | 44.06 | 68.58 | 27.02 | 32.48 |
| $\frac{1}{\log_2(1+x)}$ | $e^{2z}$ | 6.88 | 43.75 | 68.49 | **28.01** | 33.74 |
| $-\log_2(1+x)$ | $2^z$ | 6.67 | 44.84 | 69.15 | 27.96 | 33.68 |
| $(1+x)^{-1/2}$ | $2^z$ | **9.17** | 44.69 | 68.91 | 24.24 | 33.34 |
| $\frac{1}{1+x}$ | $2^z$ | **9.17** | 45.00 | 68.20 | 27.94 | 32.72 |

*Table 11.* Ablation on the final approximation used for computing $K(\xi)$ on Qwen2.5-Math-7B. We report Pass@1 and Pass@16 (higher is better). Best results within this ablation are in bold (ties are bolded).

| Model | $\xi$ Approx. | MATH-OAI | | Minerva Math | | OlympiadBench | | AIME24 | | AMC23 | |
|---|---|---|---|---|---|---|---|---|---|---|---|
| | | P@1 | P@16 | P@1 | P@16 | P@1 | P@16 | P@1 | P@16 | P@1 | P@16 |
| | Arithmetic | 66.51 | **90.20** | 32.51 | 59.19 | 31.44 | 61.93 | 5.83 | **23.33** | 37.66 | **85.00** |
| Qwen2.5-Math-7B | Geometric | 66.46 | 89.20 | 32.58 | 63.60 | 31.31 | 60.89 | **7.29** | **23.33** | 39.38 | 87.50 |
| | Logarithmic | 66.55 | **90.20** | 32.08 | **63.97** | 31.64 | 61.63 | 5.83 | 20.00 | 36.72 | 82.50 |
| | RANKTUNER (**Max**) | **68.60** | 88.80 | **33.30** | 59.56 | **32.89** | **62.07** | 7.08 | **23.33** | 44.53 | 82.50 |

few small regressions on specific benchmarks (e.g., AIME24/AMC23 for Qwen2.5-Math-1.5B and MATH-OAI Pass@1 for Qwen3-4B), which are made explicit by the $\Delta_{\text{Best}}$ rows and may reflect both benchmark-specific variance and the greater optimization difficulty of smaller-capacity backbones. Overall, RANKTUNER remains robust across architectures and datasets.

## C.5. Sensitivity to Monotone Choices of $(f, g)$

To test whether the method depends on one handcrafted transformation pair, we replace the monotone functions in Eq. (2) with several alternatives while keeping the same training and evaluation protocol on Qwen2.5-Math-7B. Tab. 10 shows that alternative monotone choices lead to broadly stable results. Different variants are best on different datasets, but the overall performance remains close, supporting that the main gain comes from the rank-based probability–entropy calibration principle rather than a specific handcrafted pair.

## C.6. Selections of $\xi$ Approximation

In RANKTUNER, $\xi$ is computed from $R$ and $\mathbb{E}[R]$. We use the **max** approximation $\xi = \max\{R, \mathbb{E}[R]\}$ by default, and compare three alternatives: (i) **Arithmetic mean**: $\xi \approx (R + \mathbb{E}[R])/2$; (ii) **Geometric mean**: $\xi \approx \sqrt{R \cdot \mathbb{E}[R]}$; (iii) **Logarithmic mean**: $\xi \approx (R - \mathbb{E}[R])/(\ln R - \ln \mathbb{E}[R])$ (with a small-difference fallback to the arithmetic mean for numerical stability). Tab. 11 reports Pass@1 and Pass@16 on five math benchmarks.

The choice of $\xi$ approximation primarily affects Pass@16, with the logarithmic and arithmetic means improving the highest-$k$ performance on MATH-OAI/Minerva Math, while the geometric mean yields the strongest Pass@1 on AIME24. Notably, the default RANKTUNER (**Max**) is robust: it achieves the best Pass@1 on MATH-OAI, Minerva Math, OlympiadBench, and AMC23, while remaining competitive at Pass@16 across benchmarks.

## C.7. Fixed versus Dynamic $K(\xi)$

To isolate whether the exponent itself should adapt to token uncertainty, we compare dynamic $K(\xi_t)$ with fixed-exponent controls on Qwen2.5-Math-7B under the same training and evaluation protocol as Sec. 5. Under this setting, the training-time dynamic $K(\xi_t)$ converges to around 0.9, making $K = 0.9$ a natural fixed control.

Tab. 12 shows that dynamic $K(\xi_t)$ outperforms all fixed-exponent controls, including the strong $K = 0.9$ control near its average training-time value. This indicates that the gain comes not only from combining probability and entropy, but also from adapting the exponent to token-level uncertainty.

*Table 12.* Fixed versus dynamic exponent ablation on Qwen2.5-Math-7B. We report Pass@1 across mathematical reasoning benchmarks; higher is better.

| Method | AIME24 | AMC23 | MATH-OAI | Minerva Math | OlympiadBench | Avg. |
|---|---|---|---|---|---|---|
| $K = 0.5$ | 5.63 | 41.09 | 66.59 | 27.71 | 32.30 | 34.66 |
| $K = 0.9$ | 4.58 | 31.09 | 56.99 | 20.80 | 23.53 | 27.40 |
| $K = 1.0$ | 2.29 | 26.88 | 53.16 | 17.44 | 18.60 | 23.67 |
| Dynamic $K(\xi_t)$ | **7.08** | **44.53** | **68.60** | **33.30** | **32.89** | **37.28** |

*Table 13.* Code generation results on HumanEval and HumanEval+. We report Pass@1 and Pass@10 (higher is better). Best results for each base model are in bold and second-best results are underlined.

| Model | Method | HumanEval | | HumanEval+ | |
|---|---|---|---|---|---|
| | | P@1 | P@10 | P@1 | P@10 |
| | Original | **51.34** | **69.05** | **41.66** | **58.62** |
| | SFT | 40.91 | 53.81 | 34.82 | 47.03 |
| Qwen2.5-Coder-3B | DFT | 36.29 | 42.93 | 31.70 | 38.13 |
| | EAFT | 39.65 | 48.78 | 34.52 | 43.87 |
| | TALR | 36.22 | 43.07 | 34.41 | 41.72 |
| | RANKTUNER ($w_t = 1$) | 41.78 | 55.31 | 35.71 | 48.70 |
| | Original | 61.06 | 77.78 | 54.49 | 71.13 |
| | SFT | 61.95 | 76.86 | 55.01 | 69.80 |
| Qwen2.5-Coder-7B | DFT | 57.40 | 69.08 | 50.65 | 63.55 |
| | EAFT | 59.37 | 70.45 | **56.10** | 70.05 |
| | TALR | 58.56 | 67.89 | 52.94 | 62.08 |
| | RANKTUNER ($w_t = 1$) | **62.72** | **78.56** | 55.76 | **71.96** |

## C.8. Code Fine-tuning and Evaluation

We study code fine-tuning on Evol-Instruct-Code-80k (Luo et al., 2023) and evaluate functional correctness on HumanEval (Chen et al., 2021) and HumanEval+ (Liu et al., 2023). We use Qwen2.5-Coder-3B and Qwen2.5-Coder-7B backbones (Hui et al., 2024). For code generation tasks, we use the general-task setting $w_t = 1$ (discussed in App. B.6) as the starting token weight. We report Pass@1 and Pass@10 (higher is better) following App. C.3.

Tab. 13 shows a clear capacity effect. On Qwen2.5-Coder-3B, fine-tuning methods can noticeably underperform the original model, suggesting that limited capacity makes it harder to absorb new code-style supervision without degrading general coding competence; under this regime, RANKTUNER is consistently the strongest fine-tuning baseline and thus best preserves performance. On the larger Qwen2.5-Coder-7B, RANKTUNER achieves the best results on three out of four metrics and remains competitive on Pass@1 of HumanEval+, suggesting that the benefits of our ranking-based scaling become more consistent as model capacity increases.

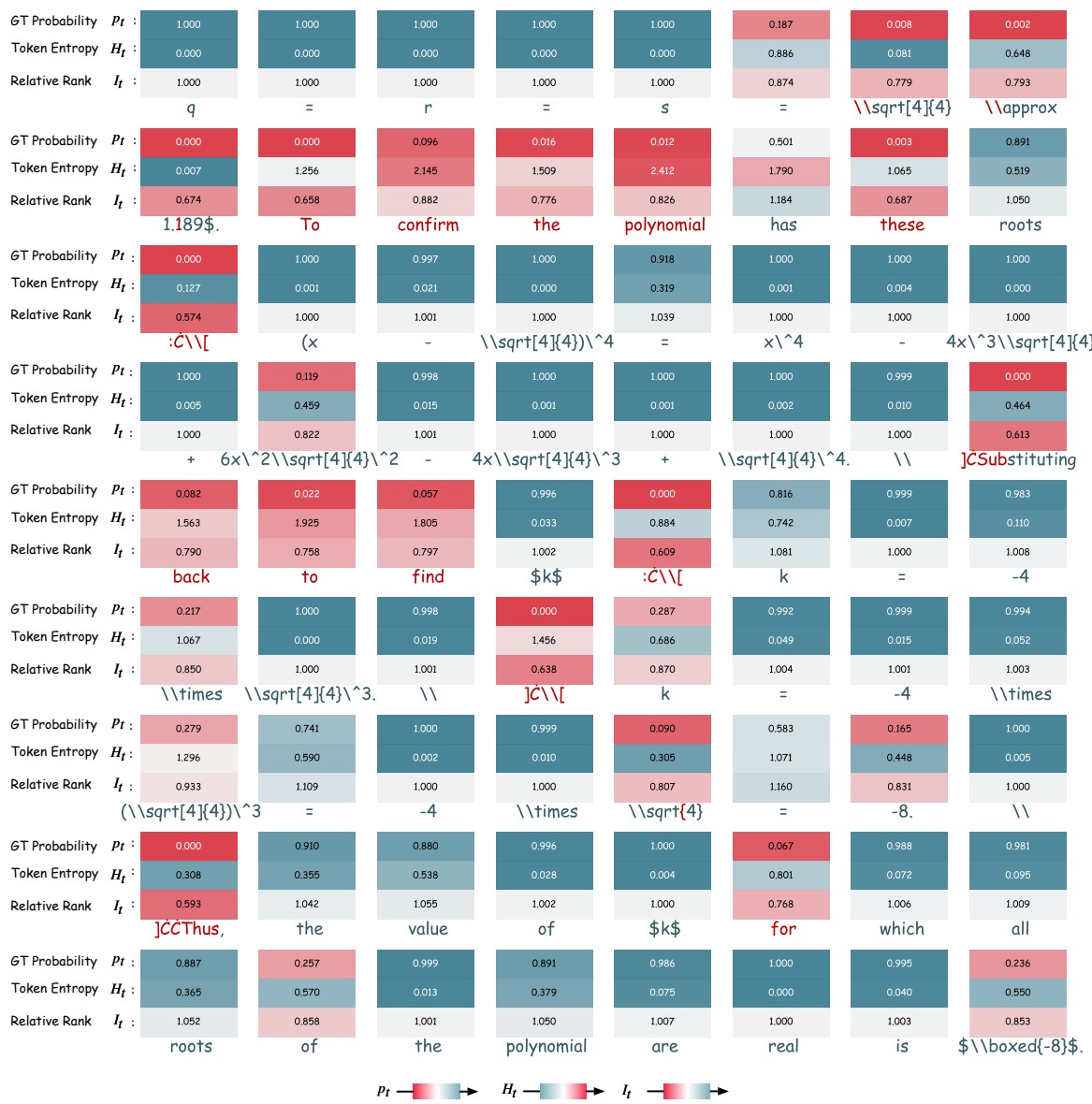

*Figure 5.* **Token-level visualization of probability, entropy, and relative-rank emphasis.** We visualize the ground-truth probability $p_t$, token entropy $H_t$, and the Relative Rank Indicator $I_t$ for one reasoning example from Numina-CoT using Qwen3-8B. The three rows correspond to $p_t$, $H_t$, and $I_t$, respectively; colors encode relative magnitude, and $I_t$ is normalized around the neutral value of 1.

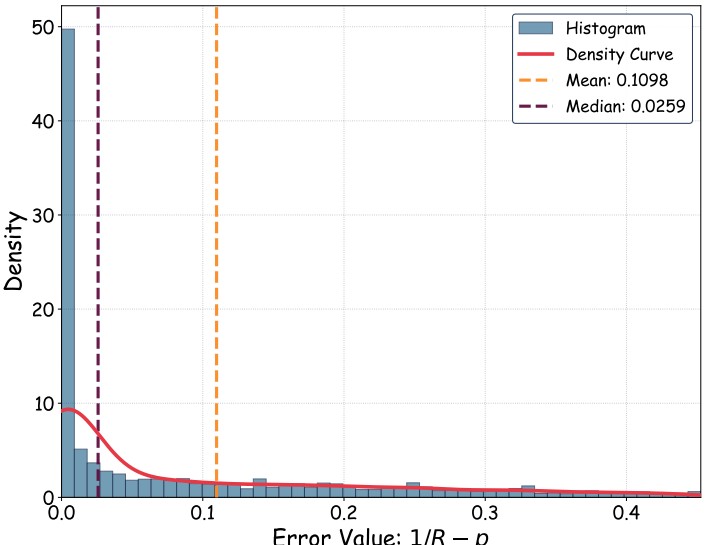

*Figure 6.* **Error distribution for the rank–probability bound on Qwen3-8B (Minerva Math, tokens 0–29).** The plot shows $\frac{1}{R} - p$, measuring how closely the rank-based surrogate $1/R$ approximates the ground-truth token probability $p$.

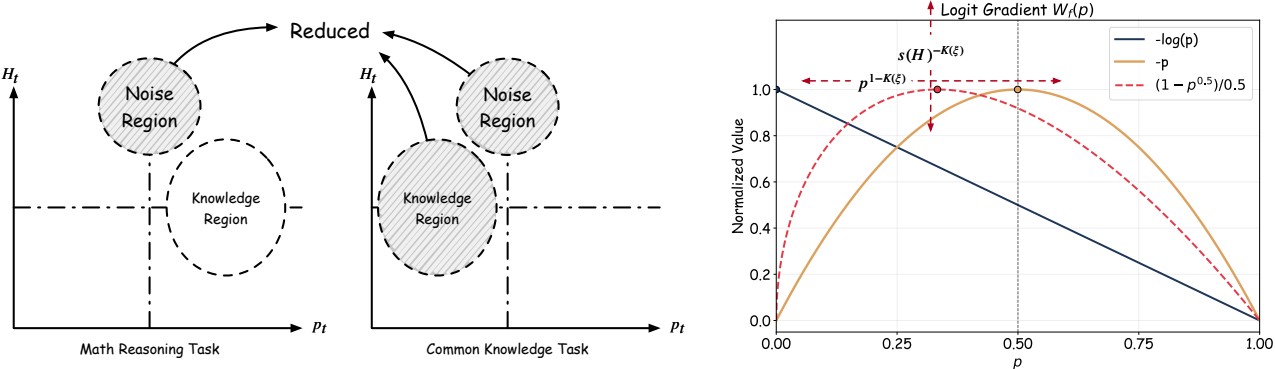

*Figure 7.* **(Left)** Illustration of the distinction between knowledge region and noise region when setting $w_t = p_t$. For math reasoning tasks, setting $w_t = p_t$ helps distinguish the knowledge region (high $p_t$) from the noise region (low $p_t$). For general tasks, if $w_t = p_t$ is applied, the knowledge region (which lies in low $p_t$) would be incorrectly delimited. **(Right)** Normalized logit-gradient magnitude $W_f(p)$ as a function of the ground-truth probability $p$ for three representative loss shapes.

