# OpenReview forum: "Probability-Entropy Calibration: An Elastic Indicator for Adaptive Fine-tuning"
_ICML.cc/2026/Conference — ICML 2026 regular_

### Official Review · Reviewer_PM2t · 2026-03-01

**Soundness:** 3
**Presentation:** 2
**Significance:** 2
**Originality:** 3
**Overall Recommendation:** 4
**Confidence:** 3

**Summary:**

The paper proposes RankTuner, a token-level reweighting method for supervised fine-tuning of LLMs. The core idea is to combine ground-truth token probability ($p_t$) and token entropy ($H_t$) into a single 'Relative Rank Indicator' ($I_t$) by comparing the actual rank of the ground-truth token against its expected rank under the model distribution. The inverse of this indicator (Relative Scale $S_t$) is used to reweight the fine-tuning loss, aiming to focus updates on genuinely under-learned tokens while down-weighting noisy or replaceable ones. Theoretical bounds connect rank-based quantities to probability and entropy via the Cauchy Mean Value Theorem. Experiments on mathematical reasoning benchmarks (MATH-OAI, Minerva Math, OlympiadBench, AIME24, AMC23) with Qwen2.5-Math-7B and Qwen3-8B show improvements over SFT and several token-reweighting baselines.

**Compliance With Llm Reviewing Policy:**

Affirmed.

**Key Questions For Authors:**

Q1. What motivates you to choose the current $f,g$ in the first place? Will a similar observation in Fig. 2(a) exist for other monotone functions

Q2. Can you explain the rationale behind defining $\xi_t$ as in Eq. 10? There could be other more explainable definitions, e.g. $\xi_t = \frac{1}{2} (\log_2 R_t +  \log_2 \mathbb{E}[R_t])$.

Q3. For the approximation K(ξ)≈0.5, do you use it as part of your empirical validation?

**Limitations:**

- Second part of Eq. 5, i.e. the relation between $\mathbb{E}[R_t]$ v.s. $p_{t, \max}$ cannot be visualize by Fig. 2. The assertion from Line 253 - 255 cannot be justified as well; see the margin between the data points and the red line in Fig.2(b). However, these are just minor flaws in presentation.

**Strengths And Weaknesses:**

Strength:
- Addresses a real problem: one-dimensional token reweighting can misidentify noisy or replaceable tokens.
- Motivated theoretical background, the starting point of the relative rank indicator is elegant. Although some flaws exist when connecting $\mathcal{I}_t$ to $H_t$ and $p_t$, the paper still provides a novel perspective.
- Statistically significant improvement over baselines on OlympiadBench P@16, AIME24 P@1,
- Ablations w/o probability/ entropy are conducted, showing the necessity of incorporating both probability and entropy under most circumstances (deteriorates the performance on pass@1 only)


Weakness:

- The motivation for defining $\hat{\kappa}$ in Eq. 8 is questionable, as most data lies in the $H_t < 2$ regime, which invalidates $\mathbb{E}[R_t] \geq \frac{1}{4} 2^{H_t} +1$; see Fig. 2(c).
- Experiments are conducted on Qwen models only. The generalizability is therefore questionable.
- Among the statistically significant improvements, only OlympicBench data has a relatively large scale, weakening the superiority of the method.
- Code generation results are less significant and the method does not work for Qwen2.5-Coder-3B

- (minor, see questions) The approximation K(ξ)≈0.5 is only valid for ξ≈1 (small ranks), but the paper applies it globally. For high-entropy tokens with large E[R], K(ξ) can be much smaller (0.076 at ξ=10), making the approximation poor.

---

> ### Author Rebuttal · Authors · 2026-03-30
>
> We thank the reviewer for the constructive feedback.
>
> > **W1. Entropy-Regime Coverage:**
> >
> Fig. 2 is only a **local visualization** of the bound, not the global entropy histogram. [**Tab. R4**](https://anonymous.4open.science/w/RanktunerRebuttalFigureTable-155C/?item=table-r4) gives full token statistics over five math benchmarks and shows the mass is **not** confined to $H_t<2$; e.g., **AMC23** has a sizable $H_t\ge2$ fraction because we analyze answer tokens only. Also, minority high-entropy tokens can still have **disproportionate training impact** [1], so entropy-aware calibration remains motivated.
>
> > **W2. Cross-Architecture Generalization:**
> >
> Appendix Table 8 already goes beyond Qwen, covering **Qwen2.5-Math-1.5B**, **Qwen3-4B**, and **Llama-3.1-8B**. The gains are consistent across architecture and scale, so the method is not supported by a Qwen-only case.
>
> > **W3. Aggregate Rather Than Single-Benchmark Gains:**
> >
> Our evidence is aggregate rather than single-dataset. On **Qwen3-8B**, RankTuner achieves the best **average performance** and **average rank** across all five math benchmarks under both Pass@1 and Pass@16. The main comparisons use **5 runs**, and the gains over **DFT** and **TALR** are statistically significant ($p<0.05$).
>
> | Method | P@1 Avg. Perf. | P@1 Avg. Rank | P@16 Avg. Perf. | P@16 Avg. Rank |
> | --- | --- | --- | --- | --- |
> | **RankTuner** | **40.73** | **1.20** | **66.26** | **1.30** |
> | DFT | 38.59 | 2.20 | 56.74 | 5.00 |
> | TALR | 39.20 | 2.60 | 63.42 | 2.50 |
> | Original | 33.08 | 4.00 | 57.74 | 4.70 |
>
> These results support **stable cross-benchmark gains**, not just one large win.
>
> > **W4. Mixed Code Results:**
> >
> We agree the code results are **more mixed**, especially on **Qwen2.5-Coder-3B**. Our claim is therefore narrower: in this lower-capacity regime, where **all** fine-tuning baselines underperform the original model, RankTuner is still the **strongest and most stable fine-tuning baseline** on `Qwen2.5-Coder-3B`, and becomes clearly more effective on **Qwen2.5-Coder-7B**.
>
> > **Q1. Monotone $f,g$ Choices:**
> >
> We use the current $f$ and $g$ because they preserve the monotone comparison between $R_t$ and $\mathbb{E}[R_t]$, keep $I_t=1$ when $R_t=\mathbb{E}[R_t]$, and yield a convenient CMVT-based power law. We do **not** claim uniqueness. [Tab. R3](https://anonymous.4open.science/w/RanktunerRebuttalFigureTable-155C/?item=table-r3) tests alternative **monotone** choices, and the behavior remains stable, suggesting the gain comes from the **rank-based probability-entropy calibration principle**, not one handcrafted pair.
>
> > **Q2. Conservative Definition of $\xi_t$:**
> >
> We define $\xi_t=\max\{R_t,\mathbb{E}[R_t]\}$ as a **conservative approximation** to the intermediate CMVT quantity; in implementation this corresponds to $\xi_t=\max\{R_t,s(H_t)\}$. Using the larger scale avoids underestimating uncertainty and is numerically robust. The appendix compares arithmetic, geometric, and logarithmic means, and the main conclusion remains stable.
>
>
> > **W5 & Q3. Large-$\xi$ Approximation Gap; Dynamic vs Fixed $K$:**
> >
> We agree that a **global** approximation $K(\xi)\approx 0.5$ is poor for large $\xi$. This is **not** used in training; it is only a small-$\xi$ interpretation from the CMVT analysis. The actual implementation always uses the **dynamic** coefficient $K(\xi_t)=\big[\log_2(\xi_t+1)\big]^{-2}$ with $\xi_t=\max\{R_t,s(H_t)\}$. [Fig. R1](https://anonymous.4open.science/w/RanktunerRebuttalFigureTable-155C/?item=figure-r1) likewise shows that the mean training-time $K(\xi)$ is dynamic rather than fixed at $0.5$.
>
> We compare the dynamic implementation against fixed-exponent controls on **Qwen2.5-Math-7B** (Pass@1):
>
> | Method | AIME24 | AMC23 | MATH-OAI | Minerva Math | OlympiadBench | Avg. |
> | --- | --- | --- | --- | --- | --- | --- |
> | $K=0.5$ | 5.63 | 41.09 | 66.59 | 27.71 | 32.30 | 34.66 |
> | $K=0.9$ | 4.58 | 31.09 | 56.99 | 20.80 | 23.53 | 27.40 |
> | $K=1.0$ | 2.29 | 26.88 | 53.16 | 17.44 | 18.60 | 23.67 |
> | **Dynamic $K(\xi)$** | **7.08** | **44.53** | **68.60** | **33.30** | **32.89** | **37.28** |
>
> The dynamic version outperforms all fixed controls on **every** benchmark.
>
> > **L1. Fig. 2 Bound Interpretation:**
> >
> We agree Fig. 2 can be visually misleading, especially for $H\ge2$. Its purpose is to illustrate the **bound relationship**, not the final indicator error. For $H<2$, the blue points are empirical $1/\mathbb{E}[R_t]$, and the paired red crosses are the upper bound $2-p_{t,\max}$. The more relevant evidence is App. B.5, where the approximation gap is quantified directly; because the final indicator applies an additional scale transformation, the visible gap in Fig. 2 does **not** translate directly into final-indicator error. We will clarify this and move the key error analysis into the main text.
>
> References:
>
> [1] Wang, Shenzhi, et al. “Beyond the 80/20 rule: High-entropy minority tokens drive effective reinforcement learning for llm reasoning.”

---

### Official Review · Reviewer_thoE · 2026-03-11

**Soundness:** 3
**Presentation:** 3
**Significance:** 3
**Originality:** 3
**Overall Recommendation:** 4
**Confidence:** 3

**Summary:**

This paper studies token-level reweighting in SFT. It argues that one-dimensional weighting can over-emphasize noisy and replaceable tokens. To address this, the paper proposes a Relative Rank Indicator that compares the target-token rank against its expected rank, yielding an uncertainty-aware adaptive token reweighting scheme for fine-tuning. Experiments on multiple base models and reasoning benchmarks show consistent improvements, and ablations support the complementary roles of probability- and entropy-aware components.

**Compliance With Llm Reviewing Policy:**

Affirmed.

**Key Questions For Authors:**

The ∆ row in Table 2 is somewhat misleading, as it reports gains over the original base model rather than over standard SFT or prior reweighting methods. This makes it difficult to isolate the actual benefit of the proposed weighting scheme from the general benefit of fine-tuning.

**Limitations:**

yes

**Strengths And Weaknesses:**

Strengths:

1. The paper addresses a real weakness of prior token-level reweighting methods for SFT, making the problem both well-motivated and practically interesting.

2. The proposed rank-based formulation provides a natural way to calibrate token importance by combining entropy and probability in a unified manner.

3. The method shows consistent improvements across multiple base models and reasoning benchmarks, while the ablations support the claimed complementary roles of probability-aware and entropy-aware components.

Weaknesses:

1. The novelty appears somewhat limited in light of recent related work such as EAFT (Entropy-Adaptive Fine-Tuning: Resolving Confident Conflicts to Mitigate Forgetting). EAFT already argues that low-probability and low-entropy “confident conflicts” should be treated differently during fine-tuning, which weakens the claim that jointly reasoning about probability and entropy is itself a new insight.

2. The literature framing may be somewhat overstated, since the proposed “prob-dominant” vs. “entropy-dominant” split does not fully capture methods like EAFT.

3. The experimental scope remains somewhat narrow. While the paper reports OOD transfer results on ARC-C and GPQA, the main training setup is still centered on 10k samples from NuminaMath-CoT, so the paper’s core conclusions remain largely tied to the mathematical reasoning setting.

---

> ### Author Rebuttal · Authors · 2026-03-30
>
> We thank the reviewer for the constructive feedback. We have narrowed our claims, clarified the difference from EAFT, and added a clearer pairwise comparison beyond the base-model $\Delta$.
>
> > **W1. Novelty vs EAFT:**
> >
> We agree that the broad intuition of combining confidence and uncertainty is **not entirely new**, and we will narrow the novelty claim accordingly. Our contribution is more specific: **RankTuner provides a rank-based calibration between target probability and contextual uncertainty**, rather than only observing that both signals matter. Relative to EAFT, the key difference is therefore not "whether entropy is used," but **how probability and entropy are coupled**.
>
> > **W2. Probability-Entropy Calibration:**
> >
> We agree the original "prob-dominant vs entropy-dominant" wording was too strong and did not fully capture EAFT. EAFT still depends on probability through the likelihood term, but its adaptive mechanism is mainly **entropy-driven**. In contrast, RankTuner treats probability as an explicit control variable and couples it with uncertainty through the Relative Rank Indicator, with scale $p_t^{-K(\xi_t)}$. This gives a more explicit and elastic calibration across task regimes, consistent with Fig. 6 (Right) and the probability-based SFT view in [1].
>
> > **W3. Beyond Math Reasoning:**
> >
> Our evidence is not limited to math reasoning: besides **OOD transfer**, **cross-architecture results**, and **code-generation** results in the appendix, we now add an *exploratory multimodal* check by fine-tuning on **WeThink** [2]. Under a setup adapted from [3] and released defaults, *RankTuner* is **best on MathVerse-Mini** and remains **competitive on MathVision**. We treat this as supportive rather than definitive evidence, but it does indicate robustness beyond a purely math-specific setting.
>
> | Method | MathVerse-Mini | MathVision |
> | --- | --- | --- |
> | Original | 33.0 | 19.9 |
> | SFT | 35.1 | 19.4 |
> | DFT | 29.8 | 16.5 |
> | EAFT | 35.2 | **20.1** |
> | TALR | 24.1 | 17.3 |
> | **RankTuner** | **36.4** | 19.6 |
>
> > **Q1. Delta Baseline Clarity:**
> >
> This is a **fair concern**. A single $\Delta$ against the Original model can blur the gain from **fine-tuning itself** and the gain from the **reweighting scheme**. To make this distinction explicit, we add [**Table R2**](https://anonymous.4open.science/w/RanktunerRebuttalFigureTable-155C/?item=table-r2), which reports the absolute results of `Original`, `SFT`, `TALR`, and *RankTuner* on **Qwen3-8B**, together with direct pairwise gains of *RankTuner* over each reference.
>
> The new comparison shows that the advantage is **not only** over the base model: averaged across benchmarks, *RankTuner* improves over **SFT** by **+15.66 P@1 / +10.23 P@16**, and over **TALR** by **+1.53 P@1 / +2.84 P@16**. Against `TALR`, it also wins or ties on **9/10** benchmark metrics. We will therefore replace the potentially misleading single-$\Delta$ presentation with clearer pairwise comparisons.
>
> References:
>
> [1] Li, Gaotang, et al. “Beyond log likelihood: Probability-based objectives for supervised fine-tuning across the model capability continuum.”
>
> [2] Yang, Jie, et al. “Wethink: Toward general-purpose vision-language reasoning via reinforcement learning.”
>
> [3] Wu, Yongliang, et al. “On the generalization of sft: A reinforcement learning perspective with reward rectification.”

---

> > ### Author Rebuttal · Reviewer_thoE · 2026-04-03
> >
> > I will keep my positive score.

---

> > > ### Author Response · Authors · 2026-04-03
> > >
> > > We sincerely appreciate your positive feedback and are pleased that our rebuttal effectively addressed your concerns. We will incorporate these points into the final version of the manuscript.

---

### Official Review · Reviewer_fGZX · 2026-03-11

**Soundness:** 3
**Presentation:** 4
**Significance:** 3
**Originality:** 3
**Overall Recommendation:** 4
**Confidence:** 3

**Summary:**

This paper presents a new weighting scheme for finetuning objectives.  Some prior work uses ground-truth token probabilities or token entropy for weighting.  This paper presents a metric that incorporates information from both measures.  Finetuning using this metric for weighting yields models that achieve higher accuracy on reasoning benchmarks.

**Compliance With Llm Reviewing Policy:**

Affirmed.

**Final Justification:**

I recommend accepting this submission.  The only reason I choose a score of 4 is that the reliance on proper assignment of the weighting parameter w_t seems to obstruct some of the generality of the technique.

**Key Questions For Authors:**

- Why do the authors make a distinction between general reasoning and math reasoning benchmarks?  They make a distinction in both the setting of $w_t$ and in the research questions they ask in their evaluation.
- I am concerned by the choice of different weighting schemes for mathematical reasoning benchmarks vs. general reasoning benchmarks.  What do the results look like if one uses the same $w_t$ for both types of benchmarks?  Is there a methodology for how to choose $w_t$ given a novel dataset?  If not, it suggests a lack of generality in the approach that should be addressed.
- Why does Section 5.4 use a different set of probability-based and entropy-based baselines than Sections 5.2 and 5.3?
- Why does Section 5.4 not include the SFT baseline?
- The weighting induced by relative scale does not intuitively seem like the correct one.  Why is it that an incorrect easy token ($p_t$ low, $H_t$ low) should achieve significantly higher weighting than an incorrect hard token ($p_t$ low, $H_t$ high)?

**Limitations:**

The main limitation I see is the different choice of $w_t$ potentially required across different datasets.

**Strengths And Weaknesses:**

**Strengths**
- The paper is technically sound, both theoretically and experimentally.
- The paper is very well written and the structure is easy to follow.
- The paper addresses a relevant, reasonably scoped problem.
- The distinction between the proposed technique and prior work is clear.

**Weaknesses**
- The motivating example in Figure 1 is not reflective of sentences one finds in reasoning datasets.
- Figure 2 shows the approximations to rank and expected rank are somewhat loose, though for the sake of performance, this is acceptable.

Overall, I found the paper enjoyable to read and made a decent contribution.  I am happy to raise my score if the questions below are resolved.

---

> ### Author Rebuttal · Authors · 2026-03-30
>
> We thank the reviewer for the careful comments and helpful suggestions.
>
> > **W1. Corpus-Faithful Example:**
> >
> We agree that Fig. 1 is mainly schematic rather than fully corpus-faithful. To address this, we now provide a **real token-level reasoning trace** in [Fig. R2](https://anonymous.4open.science/w/RanktunerRebuttalFigureTable-155C/?item=figure-r2), and will add it to the appendix. The key trend is unchanged: relatively replaceable or noisy tokens are down-weighted, while genuinely incorrect result-critical tokens receive stronger emphasis, showing how entropy helps separate ambiguity from true errors.
>
> > **W2. Approximation Tightness:**
> >
> We agree that Fig. 2, especially for $H \ge 2$, can look loose. Its purpose is to illustrate the **upper/lower-bound relationship**, not the final approximation error of the indicator itself. The more relevant evidence is the quantitative gap analysis in *App. B.5*, where we directly measure the approximation error. Because the final indicator applies an additional scale compression, the visible gap in Fig. 2 does not translate linearly into the final training signal. We will clarify this point and move the key error-analysis result into the main text.
>
> > **Q1. General vs Math Reasoning Split:**
> >
> Our main rationale follows [1]. In this view, **math reasoning** is closer to the *model-strong* regime, while **general reasoning** is closer to the *model-weak* regime. Prior work shows that model-strong tasks benefit more from probability-based weighting, corresponding to $w_t=p_t$, whereas model-weak tasks are better matched by the standard objective, corresponding to $w_t=1$. So this distinction is not an ad hoc dataset-specific choice, but a practical initialization grounded in prior SFT analysis. *RankTuner* is then applied **on top of** this starting point through the uncertainty-aware calibration term $(S_t \cdot p_t)^{-K(\xi_t)}$, whose exponent remains adaptive but bounded.
>
> > **Q2. Same $w_t$ Across Tasks; Choosing $w_t$ on New Data:**
> >
> [1] suggests a practical rule for a **new** dataset: run the base model once on the target training set and compute the **mean ground-truth probability**, which serves as a proxy for the task regime. In their Appendix B.1, *model-strong* tasks are typically around **0.75-0.80**, *model-intermediate* tasks around **0.50-0.59**, and *model-weak* tasks around **0.01-0.07**. We further verify this necessity in [Tab. R1](https://anonymous.4open.science/w/RanktunerRebuttalFigureTable-155C/?item=table-r1): across both SFT and RankTuner, switching from $w_t=1$ to $w_t=p_t$ consistently **improves math reasoning** but **hurts code generation**.
>
> > **Q3 & Q4. Section 5.4 Baselines and Missing SFT:**
> >
> Section 5.4 is an **ingredient / alternative-objective ablation**, not a second full leaderboard. Its purpose is to test whether a **single dominant dimension** is already sufficient, via `w/o Prob` and `w/o Entropy`, and whether other adaptive loss-shaping strategies, such as Alpha Power and Entropy Regularization, can match the proposed probability-entropy calibration. In this sense, the section focuses on **why RankTuner works**, while the absolute performance of the full method is already reported in **Table 2**. For the same reason, we did **not** repeat the SFT baseline in Section 5.4. For completeness, we additionally provide the full **Pass@k** comparison on **AIME24 / Qwen2.5-Math-7B** in [Fig. R3](https://anonymous.4open.science/w/RanktunerRebuttalFigureTable-155C/?item=figure-r3).
>
> > **Q5. Easy-but-Wrong vs Hard-but-Wrong Tokens:**
> >
> This is intentional. A token with **low $p_t$ and low $H_t$** means the model is wrong at a position where it should already be relatively certain, which is a stronger signal of genuine misalignment. By contrast, **low $p_t$ and high $H_t$** often indicates an intrinsically uncertain or semantically replaceable position, where multiple continuations are plausible and supervision is noisier. Over-amplifying the latter risks over-correcting ambiguous tokens. In this sense, *RankTuner measures not how wrong a token is in isolation, but how abnormal the error is relative to its uncertainty*.
>
> > **L1. Generality of the Initial $w_t$:**
> >
> We agree this can limit generality. In our method, the initial choice of $w_t$ is a **practical starting point** inherited from prior SFT findings rather than the core contribution of *RankTuner* itself. As discussed above and in App. B.6, our method is applied **on top of** that initialization. We will make this limitation explicit in the revised paper.
>
> References:
>
> [1] Li, Gaotang, et al. "Beyond log likelihood: Probability-based objectives for supervised fine-tuning across the model capability continuum."
>
> ---

---

> > ### Author Rebuttal · Reviewer_fGZX · 2026-04-03
> >
> > Most of my concerns have been addressed.  I am still quite concerned about the effects of the initial choice of w_t and why that varies across benchmarks, but I am willing to raise my score to weak accept.  The ideal evaluation for this work would sweep over both of these choices of w_t in both "general reasoning" and "math reasoning" benchmarks, but I can understand how cost constraints might prohibit that.

---

> > > ### Author Response · Authors · 2026-04-04
> > >
> > > We sincerely appreciate your positive feedback and your willingness to raise the score to a **Weak Accept**. We are glad that our previous response has addressed the **majority of your concerns**.
> > >
> > >
> > > Regarding the concern about sweeping both $w_t$ settings across both task types: The experiments in Table 2,3,4 of [1] already proved the importance to use different $w_t$ for different benchmarks. To address why the optimal $w_t$ varies across benchmarks (as discussed in Q1 & Q2): this stems from the model capability continuum [1]. Math tasks typically fall into a model-strong regime (high mean token probability $\approx 0.75\text{--}0.80$), requiring $w_t=p_t$ for precision. Coding represents a model-weak regime ($< 0.1$), where the standard $w_t=1$ is necessary to prevent over-correction.
> > >
> > >
> > > We actually also conducted some similar experiments and included them in our previous response as **[Tab. R1](https://anonymous.4open.science/w/RanktunerRebuttalFigureTable-155C/?item=table-r1)** (mentioned in response to Q2). For convenience, we have copied the table content below so you can directly view the results. The results below confirm that the choice of $w_t$ is not arbitrary but tied to the task regime: $w_t = p_t$ is essential for **Math Reasoning** (model-strong), whereas $w_t = 1$ is superior for **General Reasoning** (model-weak).
> > >
> > > 1. Math Reasoning Task
> > >
> > > | Model               | Trainer       | Setting     | AIME24     | AMC23       | MATH-OAI    | Minerva Math | OlympiadBench |
> > > | ------------------- | ------------- | ----------- | ---------- | ----------- | ----------- | ------------ | ------------- |
> > > | **Qwen3-8B**        | **SFT**       | $w_t = 1$   | 2.71       | 26.25       | 54.83       | 21.42        | 20.13         |
> > > |                     |               | $w_t = p_t$ | 8.75       | 45.78       | 70.92       | 32.42        | 35.07         |
> > > |                     |               | $\Delta$    | ↑ **6.04** | ↑ **19.53** | ↑ **16.10** | ↑ **11.01**  | ↑ **14.94**   |
> > > |                     | **RankTuner** | $w_t = 1$   | 0.83       | 14.53       | 38.25       | 11.67        | 10.40         |
> > > |                     |               | $w_t = p_t$ | 10.21      | 46.56       | 72.38       | 38.26        | 36.25         |
> > > |                     |               | $\Delta$    | ↑ **9.38** | ↑ **32.03** | ↑ **34.12** | ↑ **26.59**  | ↑ **25.85**   |
> > > | **Qwen2.5-Math-7B** | **SFT**       | $w_t = 1$   | 2.08       | 24.53       | 53.52       | 17.74        | 19.14         |
> > > |                     |               | $w_t = p_t$ | 4.17       | 41.09       | 69.15       | 26.06        | 32.62         |
> > > |                     |               | $\Delta$    | ↑ **2.08** | ↑ **16.56** | ↑ **15.62** | ↑ **8.32**   | ↑ **13.48**   |
> > > |                     | **RankTuner** | $w_t = 1$   | 0.83       | 15.78       | 41.27       | 8.66         | 9.78          |
> > > |                     |               | $w_t = p_t$ | 7.08       | 44.53       | 68.60       | 33.30        | 32.89         |
> > > |                     |               | $\Delta$    | ↑ **6.25** | ↑ **28.75** | ↑ **27.32** | ↑ **24.63**  | ↑ **23.11**   |
> > >
> > >
> > > 2. General Reasoning Task
> > >
> > > | Model                | Trainer       | Setting     | HumanEval  | HumanEval+ |
> > > | -------------------- | ------------- | ----------- | ---------- | ---------- |
> > > | **Qwen2.5-Coder-3B** | **SFT**       | $w_t = 1$   | 40.91      | 34.82      |
> > > |                      |               | $w_t = p_t$ | 36.29      | 31.70      |
> > > |                      |               | $\Delta$    | ↓ **4.62** | ↓ **3.12** |
> > > |                      | **RankTuner** | $w_t = 1$   | 41.78      | 35.71      |
> > > |                      |               | $w_t = p_t$ | 39.65      | 34.52      |
> > > |                      |               | $\Delta$    | ↓ **2.13** | ↓ **1.18** |
> > > | **Qwen2.5-Coder-7B** | **SFT**       | $w_t = 1$   | 62.72      | 55.76      |
> > > |                      |               | $w_t = p_t$ | 57.40      | 50.65      |
> > > |                      |               | $\Delta$    | ↓ **5.32** | ↓ **5.11** |
> > > |                      | **RankTuner** | $w_t = 1$   | 61.95      | 55.01      |
> > > |                      |               | $w_t = p_t$ | 59.37      | 53.04      |
> > > |                      |               | $\Delta$    | ↓ **2.59** | ↓ **1.98** |
> > >
> > >
> > >
> > > This empirical evidence supports our methodology of selecting the initial $w_t$ based on task difficulty (measured by mean ground-truth probability) before applying the **RankTuner** calibration. By grounding the initialization in this task-specific baseline, RankTuner then provides the critical second layer of uncertainty-aware calibration $(S_t \cdot p_t)^{-K(\xi_t)}$ to further optimize the training signal. We hope this experiment and its alignment with our stated methodology addresses your remaining concerns.
> > >
> > >
> > >
> > >
> > >
> > > [1] Li, Gaotang, et al. "Beyond log likelihood: Probability-based objectives for supervised fine-tuning across the model capability continuum."

---

### Official Review · Reviewer_NYzN · 2026-03-16

**Soundness:** 3
**Presentation:** 2
**Significance:** 2
**Originality:** 2
**Overall Recommendation:** 4
**Confidence:** 3

**Summary:**

This paper proposes RankTuner, a token-level reweighting method for supervised fine-tuning that aims to combine two signals usually treated separately: the probability of the ground-truth token and the entropy of the model’s predictive distribution. The main idea is to avoid overemphasizing tokens that are merely noisy or naturally replaceable, while still focusing learning on tokens that are truly under-learned. To do so, the paper introduces a Relative Rank Indicator that compares the rank of the ground-truth token against its expected rank under the model’s output distribution, and then uses its inverse as a token-wise scaling factor in the training loss. This paper reports solid gains on several math reasoning benchmarks for their method compared to other reweighting method.

**Compliance With Llm Reviewing Policy:**

Affirmed.

**Final Justification:**

Most of my concerns are addressed in the rebuttal. Thus, I would like to keep my positive recommendation.

**Key Questions For Authors:**

1. How sensitive is the method to the specific choices of $f$ and $g$?
2. Since the conclusion hints at broader training paradigms, can the authors better position RankTuner relative to RL-style post-training, or provide one comparison?
3. The paper uses Pass@1/Pass@16 in Table 2 for the math benchmarks. However, Table 10 switches to Pass@1/Pass@10 for code generation, and I did not find a clear justification for this change. Can authors provide an explanation for this difference?

**Limitations:**

No limitations are included in the submission. I would encourage authors to include a discussion about the failure cases of the proposed method.

**Strengths And Weaknesses:**

Strength:
1. The research problem is important and this paper is well motivated. The paper targets a real weakness of token reweighting and proposes an uncertainty-aware view that compares observed rank against expected rank, instead of relying on probability or entropy alone.
2. The proposed method is lightweight and easy to integrate into SFT pipeline. RankTuner introduces a clean rank-based calibration while remaining easy to integrate into weighted-NLL training, with minimal implementation overhead.
3. The proposed method presents strong empirical results on main math benchmarks and scientific reasoning benchmarks.

Weakness:
1. Theory-to-practice gap exists for the final solution. The final training weight depends on multiple approximations and stability-driven simplifications, so the method is more a principled surrogate than a tightly derived objective.
2. Ablation study is limited for the core design choice in this paper. The motivation of this paper is clear and reasonable. However, in ablation study I did not see a direct comparison between fixed K exponents (e.g., 𝐾=1 or K=0.5) and the proposed dynamic K(ξ). That makes it harder to tell how much of the gain comes from probability-entropy calibration itself versus the uncertainty-aware exponent.
3. The evaluation scope is still fairly narrow relative. Most of experiments in the paper focus on mathematic reasoning and scientific reasoning. It would also be helpful to validate the proposed method on diverse benchmarks like multi-modal reasoning benchmarks like MathVerse, MathVision and code generation benchmark like MultiPL-E.

---

> ### Author Rebuttal · Authors · 2026-03-30
>
> We thank the reviewer for the thoughtful comments and constructive suggestions.
>
> > **W1. Principled Surrogate:**
> >
> We agree. RankTuner is a **principled, stability-oriented surrogate**, not a uniquely derived optimum. The core contribution is the **rank-based calibration signal** comparing target-token alignment with intrinsic uncertainty; the approximations only make this relative-rank principle stable and usable in practice.
>
> > **W2. Fixed vs Dynamic $K$:**
> >
> We directly compare dynamic $K(\xi)$ with fixed $K=0.5$, $0.9$, and $1.0$ on **Qwen2.5-Math-7B**. Since the mean training-time $K(\xi)$ converges near $0.9$ in [Fig. R1](https://anonymous.4open.science/w/RanktunerRebuttalFigureTable-155C/?item=figure-r1), $K=0.9$ is a strong control. Dynamic $K(\xi)$ remains best, showing the gain comes not only from combining probability and entropy, but from **adapting the exponent to token uncertainty**.
>
> | Method | AIME24 | AMC23 | MATH-OAI | Minerva Math | OlympiadBench | Avg. |
> | --- | --- | --- | --- | --- | --- | --- |
> | $K=0.5$ | 5.63 | 41.09 | 66.59 | 27.71 | 32.30 | 34.66 |
> | $K=0.9$ | 4.58 | 31.09 | 56.99 | 20.80 | 23.53 | 27.40 |
> | $K=1.0$ | 2.29 | 26.88 | 53.16 | 17.44 | 18.60 | 23.67 |
> | **Dynamic $K(\xi)$** | **7.08** | **44.53** | **68.60** | **33.30** | **32.89** | **37.28** |
>
> > **W3. Broader Scope:**
> >
> We add an *exploratory multimodal* check: fine-tune on **WeThink** [1] and evaluate on **MathVerse-Mini** and **MathVision** under a setup adapted from [2] and released defaults. We treat this as supportive evidence. Even so, *RankTuner* is **best on MathVerse-Mini** and **competitive on MathVision**.
>
> | Method | MathVerse-Mini | MathVision |
> | --- | --- | --- |
> | Original | 33.0 | 19.9 |
> | SFT | 35.1 | 19.4 |
> | DFT | 29.8 | 16.5 |
> | EAFT | 35.2 | **20.1** |
> | TALR | 24.1 | 17.3 |
> | **RankTuner** | **36.4** | 19.6 |
>
> We do not add **MultiPL-E** here because its compilation-heavy evaluation stack is inconvenient to configure in our current environment.
>
> > **Q1. Sensitivity to $f,g$:**
> >
> We now provide [Tab. R3](https://anonymous.4open.science/w/RanktunerRebuttalFigureTable-155C/?item=table-r3), comparing several **monotone** choices of $f$ and $g$. Results are broadly stable: performance stays close on **AMC23**, **MATH-OAI**, and **OlympiadBench**, while different variants are best on different datasets. This suggests the gain mainly comes from the **rank-based probability-entropy calibration principle**, not one handcrafted $f,g$ pair.
>
> > **Q2. RL Relation:**
> >
> RankTuner is **complementary** to RL-style post-training. In RL, entropy is usually an **exploration regularizer**, while PPO/GRPO optimize token-level policy ratios ($\eta_t$); representative examples include *maximum-entropy RL* [3] and *ETPO* [4]. Our perspective is different: instead of rewarding entropy itself, we use it to **contextualize token difficulty**, so updates focus on tokens under-learned **relative to their uncertainty**.
>
> Two natural extensions follow: injecting our Relative Scale into PPO/GRPO updates, e.g., $\widetilde{\eta}_t(\theta)=\eta_t(\theta)\cdot\mathcal{S}_t$, or using relative rank / $\mathcal{S}_t$ for token selection, in the spirit of [5]. We do **not** claim these RL variants are validated here.
>
> > **Q3. Pass@k:**
> >
> We follow standard protocols: **Pass@1/Pass@16** for math reasoning and **Pass@1/Pass@10** for code generation [6]. The choice is also motivated by task characteristics:
>
> - **Code:** $n=50$, $temp=0.2$, $top_p=0.95$, report **Pass@1/Pass@10**
> - **Math:** $n=16$, $temp=1.0$, $top_p=1.0$, report **Pass@1/Pass@16**
>
> Code usually benefits from lower-temperature sampling with strict executable verification, while math benefits from higher-temperature sampling and more diverse valid traces. Across both settings, **Pass@1** measures single-sample precision and **Pass@10/16** multi-sample coverage.
>
> > **L1. Limits:**
> >
> Our strongest evidence is in the current SFT setting; extensions to RL and broader tasks still require validation. The main claim here is therefore the **calibrated indicator itself**, which may also inform future *token-*, *step-*, and *sequence-level* importance design.
>
> References:
>
> [1] Yang, Jie, et al. "Wethink: Toward general-purpose vision-language reasoning via reinforcement learning."
>
> [2] Wu, Yongliang, et al. "On the generalization of sft: A reinforcement learning perspective with reward rectification."
>
> [3] Haarnoja, Tuomas, et al. "Soft actor-critic: Off-policy maximum entropy deep reinforcement learning with a stochastic actor."
>
> [4] Wen, Muning, et al. "Entropy-regularized token-level policy optimization for language agent reinforcement."
>
> [5] Wang, Shenzhi, et al. "Beyond the 80/20 rule: High-entropy minority tokens drive effective reinforcement learning for llm reasoning."
>
> [6] Chen, Mark, et al. "Evaluating large language models trained on code."

---

> > ### Author Rebuttal · Reviewer_NYzN · 2026-04-03
> >
> > Most of my concerns are addressed. I would encourage authors to include a more comprehensive discussion to RL-style post-training the revision to reduce the confusion.

---

> > > ### Author Response · Authors · 2026-04-03
> > >
> > > We appreciate your positive feedback and suggestions and are pleased that our rebuttal addressed your most concerns. We will broaden our method via exploring its application on classic RLVR method on the next step!

---

### Decision · Program_Chairs · 2026-04-30

**Decision:**

Accept (regular)

**Comment:**

This is a solid well written paper that takes a principled approach for reweighting tokens during SFT. It proposes a new metric that combines ground-truth probability with entropy to create the weight function.  The method is easy is to integrate into an existing SFT pipeline. From the rebuttal it is clear that the authors have thought through many aspects of the research problem. For example, when a reviewer asked what is the point of separating math-reasoning vs. reasoning tasks, the authors provided a nuanced answer grounded in previous work - distinction between strong vs. weak model regime.